# TOKEN-GUARD: TOWARDS TOKEN-LEVEL HALLUCINATION CONTROL VIA SELF-CHECKING DECODING

**Yifan Zhu[1], Huiqiang Rong[1], Haoran Luo[2]***
[1]Beijing University of Posts and Telecommunications  [2]Nanyang Technological University
{yifan_zhu,rhq}@bupt.edu.cn, haoran.luo@ieee.org

## ABSTRACT

Large Language Models (LLMs) often hallucinate, generating content inconsistent with the input. Retrieval-Augmented Generation (RAG) and Reinforcement Learning with Human Feedback (RLHF) can mitigate hallucinations but require resource-intensive retrieval or large-scale fine-tuning. Decoding-based methods are lighter yet lack explicit hallucination control. To address this, we present **Token-Guard**, a token-level hallucination control method based on self-checking decoding. Token-Guard performs internal verification at each reasoning step to detect hallucinated tokens before they propagate. Candidate fragments are further evaluated in a latent space with explicit hallucination risk scoring, while iterative pruning and regeneration dynamically correct detected errors. Experiments on HALU datasets show Token-Guard substantially reduces hallucinations and improves generation accuracy, offering a scalable, modular solution for reliable LLM outputs. Our code is publicly available[1].

## 1 INTRODUCTION

Large Language Models (LLMs) (Zhao et al., 2025) have achieved widespread success in natural language processing tasks. However, in knowledge-intensive or proprietary scenarios, they still suffer from the hallucination problem (Zhang et al., 2023), generating inaccurate or misleading content. To improve factual consistency, strategies such as RAG (Lewis et al., 2020; Luo et al., 2025b) and RLHF (Stiennon et al., 2020) have been proposed. These methods alleviate hallucinations but heavily rely on external knowledge retrieval or large-scale fine-tuning, which is resource-intensive. To address this, decoding-based methods have been adopted to further enhance the model's generation process, improving both quality and output reliability.

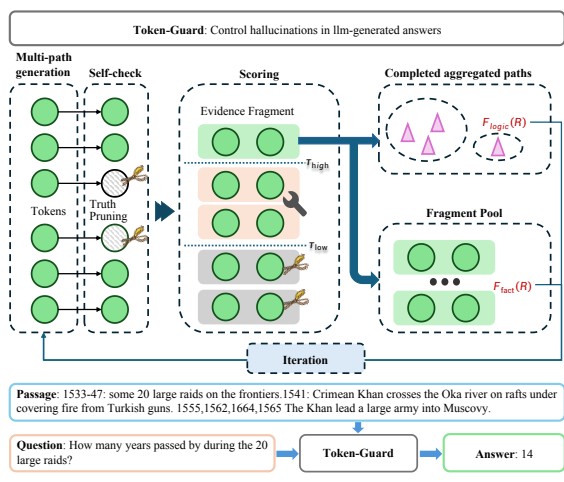

Figure 1: An illustration of Token-Guard.

Existing decoding methods, including Auto-Regressive Chain-of-Thought(CoT) (Wei et al., 2022), Tree-of-Thought (Yao et al., 2023), Guided (Xie et al., 2023), and Predictive Decoding (Ma et al., 2024), share three main characteristics: First, outputs are generated progressively through step-wise reasoning, allowing the model to build multi-step chains of thought. Second, exploratory generation is often adopted, where multiple reasoning paths are expanded in parallel to increase the chance of reaching a correct solution. Finally, intermediate self-evaluation provides feedback at each reasoning step, enabling partial assessment of coherence and consistency before proceeding further.

---

* Corresponding author.
[1]https://github.com/rhq945/Token-Guard

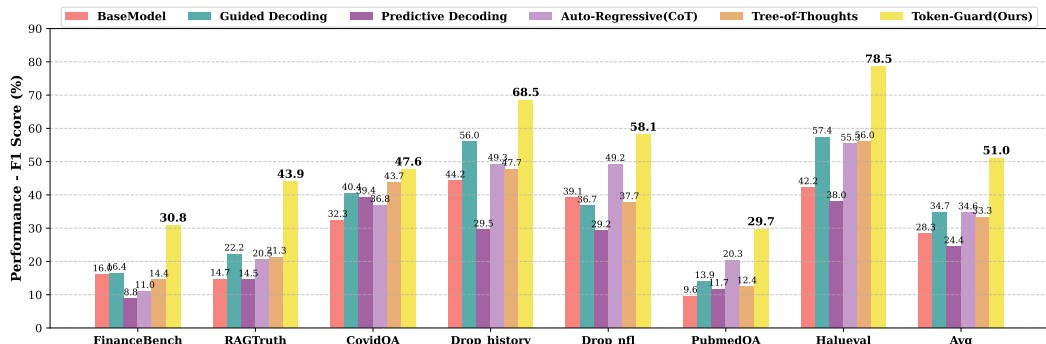

Figure 2: Comparison of F1 scores across HALU benchmarks, illustrating model performance on various tasks and effectiveness in hallucination mitigation.

Despite these advances, current decoding methods still face three major challenges in hallucination control: **(i) Limited token-level hallucination checking**. Most existing decoding methods do not implement an explicit token-level hallucination checking mechanism, relying instead on general representations without self-validation. This may allow token-level errors in reasoning steps to propagate and compound (Zhang et al., 2025). **(ii) Hallucination risk not explicitly quantified.** Traditional probability scoring methods fail to explicitly quantify hallucination risk in the encoded space, leading to undirected token selection and unstable reasoning results (Zeng et al., 2024). **(iii) Limited dynamic correction capability in iterative and generation stages.** Most methods support only single-pass generation or limited iterations, lacking mechanisms for dynamic correction and resource allocation, which undermines output consistency and may incur unnecessary time and computational costs (Ugare et al., 2024).

To address these challenges, we propose `Token-Guard` (see Figure 1), a token-level hallucination-controlled decoding method. Token-Guard introduces three key innovations: **(1) Token-level Hallucination Control.** At each reasoning step, we score candidate tokens in the latent space and prune low-confidence tokens, enabling the model to effectively detect and suppress hallucinated tokens, improving factual precision. **(2) Segment-level Explicit Hallucination Scoring.** We group relevant tokens into candidate fragments and assign hallucination risk scores, guiding more reliable and accurate fragment selection, thus improving relevance. **(3) Local Enhancement and Global Iteration.** When hallucinations are detected, local fragment regeneration and global iteration strategies are applied to dynamically correct prior fragments. For multi-fragment reasoning paths, these strategies maintain logical consistency while avoiding unnecessary time and computational overhead.

We evaluated Token-Guard on multiple standard HALU datasets. Experimental results show that Token-Guard, when integrated with llm, achieves substantial improvements over both conventional decoding-based approaches combined with large models and the base large models themselves. Specifically, Token-Guard demonstrates significant gains in generation accuracy, with relative improvements of up to 16.3% over the strongest baseline, and notable enhancements in output quality, including logical and factual consistency, as illustrated in Figure 2.

## 2 RELATED WORK

**LLM Hallucination Control.** LLMs often produce hallucinations—outputs that are factually incorrect or unsupported. Existing approaches address this through RAG (Lewis et al., 2020; Shuster et al., 2021; Luo et al., 2025a) with domain-specific pipelines (Li et al., 2024) and benchmarks like RAGTruth (Wu et al., 2023), alignment methods including instruction fine-tuning and RLHF (Christiano et al., 2017; Stiennon et al., 2020; Bai et al., 2022) with multi-modal or interactive extensions (Gunjal et al., 2023; Al-Yaychli et al., 2025), and post-generation detection via uncertainty estimation (Ledger & Mancinni, 2024) or classifiers like HaloScope (Du et al., 2024). These methods can be computationally intensive or domain-specific, limiting practical use.

**Decoding and Decoding Methods for Large Language Models.** Decoding strategies significantly improve reasoning quality and efficiency. Chain-of-Thought (CoT) enables explicit multi-step reasoning. Methods such as DoLa (Chuang et al., 2024) and KCTS (Choi et al., 2023) introduce dif-

ferent forms of token-level hallucination detection, allowing models to filter unreliable tokens and enhance factual consistency during generation. Phi-Decoding (Xu et al., 2025) uses adaptive pruning and foresight sampling to improve computational efficiency and better control hallucinations. Building on multi-level control (Hiriyanna & Zhao, 2025) and self-reflective decoding (Ji et al., 2023b), we introduce Token-Guard, a decoding method designed to provide **explicit and robust hallucination reduction** throughout the LLM reasoning process.

## 3 PRELIMINARIES

We summarize key concepts and fundamental principles underlying token-level generation, multi-level supervision, and iterative verification processes in modern autoregressive models:

**(a) Token-Level Auto-Regressive Generation.** A sequence of tokens $a_1, \ldots, a_T$ is generated autoregressively. Formally, the probability of the entire sequence given input $x$ is defined as:

$$P(a_1, \ldots, a_T \mid x) = \prod_{t=1}^{T} P_\theta(a_t \mid x, a_{<t}), \tag{1}$$

where $T$ is the sequence length, $a_{<t} = \{a_1, \ldots, a_{t-1}\}$ denotes the tokens generated before step $t$, and $P_\theta$ represents the model-assigned conditional probability of $a_t$.

**(b) Multi-Level Supervision.** Token probabilities can be modulated by hierarchical signals:

$$C_t = \sum_{l \in \mathcal{L}} w_l \, g_l(a_t \mid x, a_{<t}, C_{t-1}), \tag{2}$$

where $\mathcal{L}$ is the set of supervision levels, $g_l$ is the control function at level $l$, $w_l$ is its weight, and $C_{t-1}$ is the previous combined score. $C_t$ represents a composite guidance score.

**(c) Iterative Verification.** Generated sequences can be refined iteratively:

$$K^{(t+1)} = \begin{cases} K^{(t)}, & S(K^{(t)} \mid Q) \geq \tau \\ R(K^{(t)}, S(K^{(t)} \mid Q), Q), & S(K^{(t)} \mid Q) < \tau \end{cases} \tag{3}$$

where $K^{(t)}$ is the sequence at iteration $t$, $Q$ is the input query, $S(\cdot \mid Q)$ is a score measuring correctness or hallucination risk, $\tau$ is the threshold, and $R(\cdot)$ is a refinement function applied to low-scoring sequences to improve accuracy and reduce potential hallucinations.

## 4 METHODOLOGY: TOKEN-GUARD

In this section, as illustrated in Figure 3, we introduce Token-Guard, including token-level self-checking initialization, iterative token scoring and local fix, pruning-based candidate selection, global iratation and final response generation to control hallucinations during decoding.

### 4.1 PROMPT SELF-CHECKING

Before the Token-Guard pipeline, we use a unified prompting strategy combining a general template with domain-specific constraints. The general prompt enforces reliance on the input and a standardized output format ("Answer:[...]"), while domain-specific prompts add restrictions such as "Yes/No/Maybe" in **PubMedQA**, financial reporting in **FinanceBench**, or input-dependent answers in **History**, **CovidQA**, and **RagTruth**. Full details are provided in the Appendix A.1.

### 4.2 TOKEN-LEVEL HALLUCINATION SELF-CHECKING

Token-Guard establishes a controllable generation space $\mathcal{S}_0$ and performs token-level self-checking to reduce hallucination risks. Each token $a_t$ is verified before being propagated, limiting local hallucinations and supporting multi-step reasoning.

**Latent Token Environment Initialization.** At generation step $t$, a latent environment $S_t$ is constructed to store semantic representations $s_j$ and contextual states $h_j$ of accepted tokens $a_j$ for $j < t$.

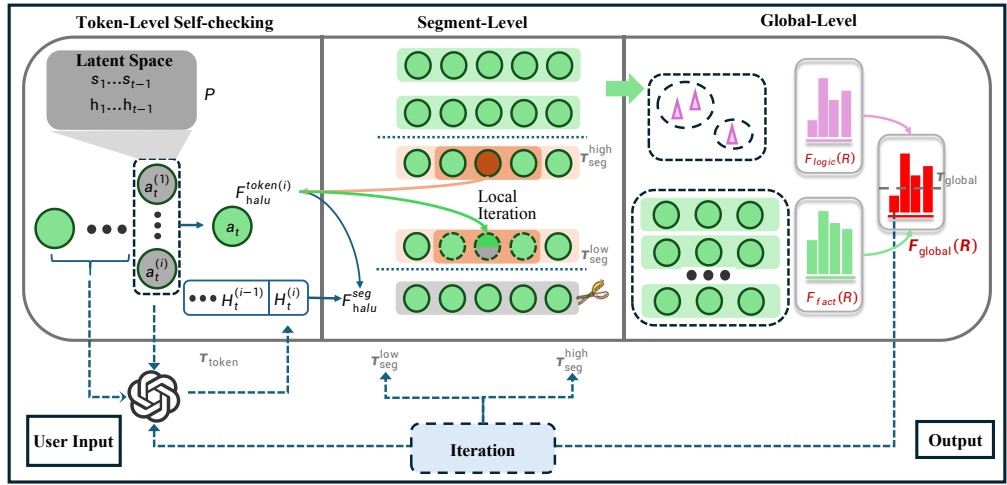

Figure 3: Overview of the Token-Guard framework: an iterative token-level decoding trajectory with self-checking, hallucination scoring, local fix, and pruning to ensure reliable output.

To address the initialization of the latent environment at $t = 1$, we define the mean hidden state of the input context tokens as an initial anchor:

$$h_x := \frac{1}{|x|} \sum_{i=1}^{|x|} \text{LLM}_{\text{hidden}}^{(L-1)}(x_i), \quad \bar{h}_{<1} := h_x. \tag{4}$$

where $x = (x_1, \ldots, x_{|x|})$ denotes the input context token and $\text{LLM}_{\text{hidden}}^{(L-1)}$ denotes the penultimate-layer hidden representation produced by the model. This ensures that trajectory coherence for the first candidate token is well-defined and that hallucination scoring is applicable from the start.

**Candidate Tokens and Hidden States.** The LLM generates a candidate token set $\mathcal{A}_t = \{a_t^{(1)}, \ldots, a_t^{(M)}\}$, each with a hidden state $h_t^{(i)}$ defined by:

$$h_t^{(i)} = \text{LLM}_{\text{hidden}}^{(L-1)}\big(a_t^{(i)}, a_{<t}, x\big), \tag{5}$$

where $a_{<t}$ is the sequence of previously accepted tokens and $x$ denotes the input context.

**Hybrid Token-Level Hallucination Scoring.** For each candidate token $a_t^{(i)}$, we first compute the mean hidden state of previously accepted tokens: $\bar{h}_{<t} = \frac{1}{t-1} \sum_{j=1}^{t-1} h_j, t > 1$, where $h_j$ is the hidden state of token $a_j$. Then, the hybrid token-level hallucination score is then defined as

$$F_{\text{halu}}^{\text{token}}(a_t^{(i)} \mid s_t) = \lambda \cdot \frac{h_t^{(i)} \cdot \bar{h}_{<t}}{|h_t^{(i)}| |\bar{h}_{<t}|} + (1 - \lambda) \cdot P(a_t^{(i)} \mid a_{<t}, x), \tag{6}$$

where $\lambda \in [0, 1]$ balances semantic consistency and token probability. The weighted combination yields an interpretable score and supports easy tuning of the semantic–probabilistic trade-off. In our experiments, we set $\lambda = 0.6$.

**Candidate Token Selection and Environment Update.** Select the next token $a_t^*$ from the candidate set $\mathcal{A}_t$ by

$$a_t^* = \arg\max_{i \in \mathcal{A}_t} \mathbf{1}\{F_{\text{halu}}^{\text{token}}(a_t^{(i)} \mid s_t) \geq \tau_{\text{token}}\}, \qquad s_{t+1} = s_t \cup \{h_t^{(a_t^*)}\}. \tag{7}$$

Consecutive tokens passing the token-level threshold form candidate segments $\mathcal{C}_k$, preserving semantic coherence for segment-level scoring. In our experiments, we set $\tau_{\text{token}} = 0.4$.

*Memory note:* Only a running average $\bar{h}_{<t}$ is maintained for scoring; token hidden states are temporarily buffered until segment formation, then released, keeping memory $\mathcal{O}(L_{\max} \cdot K_{\text{active}} \cdot d)$ independent of total generation length $T$.

**Proposition 1.** *Token-level self-checking reduces expected hallucination of generated sequences.*

*Proof.* Experimental results are provided in Section 5.4 and theoretical proofs in Appendix B.1. □

### 4.3 CANDIDATE SEGMENT REPRESENTATION

Let a candidate segment $C_k$ consist of a sequence of self-checked tokens $\{a_{t_1}, \ldots, a_{t_n}\}$, where each token $a_{t_i}$ has an associated hidden state $h_t^{(i)}$ obtained from the token-level self-checking process. We construct a segment-level representation $H_k$ as a weighted average of these token hidden states, where the weights $w_i$ are derived from the token-level hallucination scores $F_{\text{halu}}^{\text{token}}(a_{t_i} \mid s_{t_i})$ using a softmax function to emphasize more reliable tokens:

$$w_i = \frac{\exp(F_{\text{halu}}^{\text{token}}(a_{t_i} \mid s_{t_i}))}{\sum_{j=1}^{n} \exp(F_{\text{halu}}^{\text{token}}(a_{t_j} \mid s_{t_j}))}, \quad H_k = \sum_{i=1}^{n} w_i h_t^{(i)}. \tag{8}$$

This ensures tokens with higher reliability contribute more to $H_k$, capturing the most trustworthy semantic content. Each segment $C_k$ is then evaluated using three complementary aspects. First, the weighted token-level hallucination score aggregates individual token reliabilities:

$$F_{\text{halu}}^{\text{token}}(C_k) = \sum_{i=1}^{n} w_i F_{\text{halu}}^{\text{token}}(a_{t_i} \mid s_{t_i}). \tag{9}$$

Second, local consistency measures the smoothness of semantic transitions between adjacent tokens, with abrupt changes potentially indicating hallucinations or logical breaks:

$$\text{Consistency}(C_k) = 1 - \frac{1}{n-1} \sum_{i=1}^{n-1} |h_t^{(i)} - h_t^{(i+1)}|. \tag{10}$$

Third, global alignment evaluates how well the segment aligns with the overall input context representation $H_x$, computed as the cosine similarity between the segment representation $H_k$ and $H_x$:

$$\text{Alignment}(C_k) = \frac{H_k \cdot H_x}{|H_k| \, |H_x|}. \tag{11}$$

The segment-level hallucination score is computed by a weighted sum of these three components:

$$F_{\text{halu}}^{\text{seg}}(C_k) = \alpha F_{\text{halu}}^{\text{token}}(C_k) + \beta \, \text{Consistency}(C_k) + \gamma \, \text{Alignment}(C_k), \tag{12}$$

where $\alpha, \beta, \gamma \in [0, 1]$ represent the relative importance of the three parts, respectively, satisfying $\alpha + \beta + \gamma = 1$. In our experiments, we set $\alpha = 0.5, \beta = 0.3, \gamma = 0.2$.

Segments above $\tau_{\text{seg}}^{\text{high}}$ are valid and stored or used in multi-step reasoning. Segments between $\tau_{\text{seg}}^{\text{low}}$ and $\tau_{\text{seg}}^{\text{high}}$ are refined, while those below $\tau_{\text{seg}}^{\text{low}}$ are discarded to keep only reliable, coherent content.

During local refinement, only the target segment $C_k$ is updated. The hidden states of downstream segments $C_{k+1}, C_{k+2}, \ldots$ are retained to preserve global structure, avoiding full regeneration of the remaining sequence and ensuring efficiency, following the principle of selective local refinement as demonstrated in (Tang et al., 2025; Lyu et al., 2025).

For local refinement, the lowest-scoring token $a_{\text{low}}$ is identified within the segment, and a local window $W_k^{(l)} = \{a_{i-1}, a_i^{\text{low}}, a_{i+1}\}$ is formed including its immediate neighbors. This window is used to correct the weakest part of the segment while preserving surrounding context. The LLM generates refined tokens conditioned on the surrounding context $a_{<i-1}, a_{>i+1}$ and the segment representation $H_k$, producing a refined window $W_k^{(l)'}$ that replaces the original window in the segment:

$$W_k^{(l)'} = \text{LLM\_refine}(W_k^{(l)} \mid a_{<i-1}, a_{>i+1}, H_k), \quad C_k' = C_k \setminus W_k^{(l)} \cup W_k^{(l)'}. \tag{13}$$

After refinement, the segment score $F_{\text{halu}}^{\text{seg}}(C_k')$ is recomputed. If it reaches $\tau_{\text{seg}}^{\text{high}}$, the segment is accepted; otherwise, iteration continues up to $N_{\max}$ steps. This process ensures a controlled, step-wise improvement of segment reliability. We set $\tau_{\text{seg}}^{\text{low}} = 0.55, \tau_{\text{seg}}^{\text{high}} = 0.75, N_{\max} = 3$.

*Memory note:* During segment formation and local refinement, token hidden states within the segment are temporarily buffered. Once the final segment representation $H_k$ is computed, temporary hidden states are released. Downstream segment vectors are retained only as compact representations, bounding memory to $\mathcal{O}(L_{\max} \cdot d + K \cdot d)$, independent of total generation length $T$.

**Proposition 2.** *Segment-level with local refinement improves quality and reduces hallucinations.*

*Proof.* We provide experimental results in Section 5.5 and theoretical proofs in Appendix B.2. $\square$

## 4.4 GLOBAL ITERATION AND CORRECTION

In Stage Three, Token-Guard globally re-evaluates candidate outputs from Stage Two by assembling reliable segments into reasoning chains $R = \{C_1, C_2, \ldots, C_K\}$. We do not rely on the original segment order, since exploring multiple branches can improve global consistency and factual accuracy, inspired by Saha et al. (2024). Segments are clustered via TF-IDF and KMeans, and candidate chains are generated by selecting the chain closest to each cluster centroid. We set the number of clusters to $K = 5$ for large-scale datasets, while for small datasets such as HaluEval we use $K = 3$ to reduce computation. Each segment $C_k$ has a vector representation $H_k$, a token-level confidence vector $\mathbf{f}_k$, and a knowledge-verification score $E_k$ comparing generated content with relevant knowledge sources. The factual consistency of a chain is computed as:

$$F_{\text{fact}}(R) = \frac{1}{K} \sum_{k=1}^{K} w_k \, F_{\text{seg}}(C_k), \quad w_k = \frac{\|\mathbf{f}_k\| \cdot E_k}{\sum_{j=1}^{K} \|\mathbf{f}_j\| \cdot E_j}, \tag{14}$$

where $F_{\text{seg}}(C_k)$ means segment-level hallucination score, and $w_k$ combines token confidence and evidence alignment to emphasize longer segments. Logical coherence is measured between adjacent segments using cosine similarity of $H_k$ and $H_{k+1}$, weighted by contextual factor $\lambda_k$:

$$F_{\text{logic}}(R) = \frac{1}{K-1} \sum_{k=1}^{K-1} \lambda_k \frac{H_k \cdot H_{k+1}}{\|H_k\| \, \|H_{k+1}\|}, \quad \lambda_k = \text{sim}_{\text{ctx}}(C_k, C_{k+1}), \tag{15}$$

where $\text{sim}_{\text{ctx}}$ provides a multiplicative contextual adjustment and is defined as:

$$\text{sim}_{\text{ctx}}(C_k, C_{k+1}) = \frac{1 + \cos(\tilde{e}_k, \tilde{e}_{k+1})}{2}, \quad \tilde{e}_k = \frac{1}{n_k} \sum_{i=1}^{n_k} e_{k,i}, \tag{16}$$

with $e_{k,i}$ denoting the input token embedding of segment $C_k$. This factorized formulation combines hidden-state continuity through $\cos(H_k, H_{k+1})$ and semantic alignment between segment texts through $\text{sim}_{\text{ctx}}$, which uses only input embeddings reconstructed from token IDs.

The global score $F_{\text{global}}(R)$ combines factual and logical components via a soft minimum:

$$F_{\text{global}}(R) = \frac{F_{\text{fact}}(R) \, F_{\text{logic}}(R)}{F_{\text{fact}}(R) + F_{\text{logic}}(R) - F_{\text{fact}}(R) \, F_{\text{logic}}(R)}, \tag{17}$$

triggering re-generation if $F_{\text{global}}(R) < \tau_{\text{global}}$, which we set 0.7, and returning "cannot answer" if both $F_{\text{fact}}$ and $F_{\text{logic}}$ are below 0.5. To improve stability, segment thresholds are dynamically adjusted by an adjustment margin $\Delta\tau$:

$$\tau_{\text{seg}}^{\text{adj}} = \begin{cases} \tau_{\text{seg}}^{\text{high}} + \Delta\tau, & \text{if } F_{\text{fact}} \text{ is low and } F_{\text{logic}} \text{ is high}, \\ \tau_{\text{seg}}^{\text{low}} - \Delta\tau, & \text{if } F_{\text{logic}} \text{ is low and } F_{\text{fact}} \text{ is high}. \end{cases} \tag{18}$$

where $\tau_{\text{seg}}^{\text{adj}}$ is the adjusted threshold. Each iteration produces a candidate chain $R^{(i)}$ and recalculates $F_{\text{global}}^{(i)}$; the chain is accepted if $F_{\text{global}}^{(i)} \geq \tau_{\text{global}}$ before $M_{\text{max}}$ iterations, otherwise the system outputs "cannot answer." In our experience, we set $\tau_{\text{global}} = 0.7, \Delta\tau = 0.1, M_{\text{max}} = 2$

*Memory note:* Global alignment and scoring operate exclusively on compact segment vectors $\{H_k\}_{k=1}^{K}$ and token-level confidence vectors $\mathbf{f}_k$. No full token hidden sequences are stored, keeping memory usage bounded by $\mathcal{O}(K \cdot d)$, independent of total generation length $T$.

**Proposition 3.** *Global iterative refinement over candidate sequences reduces hallucination.*

*Proof.* We provide experimental results in Section 5.6 and theoretical proofs in Appendix B.3. □

## 5 EXPERIMENTS

This section presents the experimental setup, main results, and analysis. We answer the following research questions (RQs): **RQ1:** Does Token-Guard outperform other methods? **RQ2:** Does the main component of Token-Guard work, and how is its comparative analysis? **RQ3–5:** How well does **Token-Guard** improve the overall generation quality, in terms of factual precision and correctness, relevance to the query, and logical consistency and completeness in multi-step reasoning? **RQ6:** How does Token-Guard affect time efficiency and memory consumption during decoding? **RQ7:** How does Token-Guard perform across backbone models of different scales?

| Method | FinanceBench | | RAGTruth | | CovidQA | | DROP_history | | DROP_nfl | | PubmedQA | | Halueval | | Avg | |
|---|---|---|---|---|---|---|---|---|---|---|---|---|---|---|---|---|
| | EM | F1 | EM | F1 | EM | F1 | EM | F1 | EM | F1 | EM | F1 | EM | F1 | EM | F1 |
| *Meta-Llama-3.1-8B-Instruct* | | | | | | | | | | | | | | | | |
| BaseModel | 0.16 | 16.00 | 0.00 | 14.71 | 0.02 | 32.33 | 0.24 | 44.21 | 0.30 | 39.10 | 0.00 | 9.55 | 0.32 | 42.16 | 0.15 | 28.29 |
| Guided Decoding | 0.14 | 16.44 | 0.00 | 22.21 | 0.04 | 40.43 | 0.34 | 55.95 | 0.18 | 36.71 | 0.00 | 13.95 | 0.42 | 57.41 | 0.16 | 34.73 |
| Predictive Decoding | 0.09 | 8.79 | 0.00 | 14.48 | 0.04 | 39.36 | 0.14 | 29.47 | 0.20 | 29.22 | 0.00 | 11.69 | 0.22 | 38.00 | 0.10 | 24.43 |
| Chain-of-Thoughts | 0.11 | 11.01 | 0.00 | 20.47 | 0.08 | 36.84 | 0.30 | 49.26 | 0.34 | 49.21 | 0.00 | 20.33 | 0.40 | 55.32 | 0.18 | 34.63 |
| Tree-of-Thought | 0.10 | 14.44 | 0.00 | 21.33 | 0.10 | 43.70 | 0.22 | 47.73 | 0.24 | 37.69 | 0.00 | 12.38 | 0.38 | 56.02 | 0.15 | 33.33 |
| **Token-Guard(Ours))** | **0.30** | **30.80** | **0.02** | **43.94** | **0.08** | **47.64** | **0.48** | **68.52** | **0.44** | **58.10** | **0.00** | **29.67** | **0.68** | **78.54** | **0.29** | **51.03** |
| *Qwen3-8b* | | | | | | | | | | | | | | | | |
| BaseModel | 0.20 | 26.67 | 0.00 | 36.12 | 0.05 | 42.57 | 0.44 | 65.10 | 0.39 | 57.02 | 0.00 | 22.45 | 0.43 | 59.83 | 0.22 | 44.25 |
| Guided Decoding | 0.21 | 23.56 | 0.00 | 39.35 | 0.07 | 43.19 | 0.52 | 66.11 | 0.43 | 53.68 | 0.00 | 24.76 | 0.62 | 69.88 | 0.26 | 45.79 |
| Predictive Decoding | 0.04 | 11.25 | 0.00 | 34.51 | 0.00 | 31.92 | 0.06 | 23.85 | 0.00 | 23.41 | 0.00 | 19.44 | 0.00 | 21.29 | 0.01 | 23.67 |
| Chain-of-Thoughts | 0.29 | 35.12 | 0.00 | 33.78 | 0.00 | 41.63 | 0.48 | 69.33 | 0.49 | 59.89 | 0.00 | 22.77 | 0.34 | 53.21 | 0.23 | 45.10 |
| Tree-of-Thought | 0.28 | 34.91 | 0.00 | 36.07 | 0.00 | 37.34 | 0.49 | 69.12 | 0.34 | 54.67 | 0.00 | 26.41 | 0.39 | 57.33 | 0.21 | 45.12 |
| **Token-Guard(Ours)** | **0.45** | **45.37** | **0.06** | **45.89** | **0.09** | **44.01** | **0.66** | **71.83** | **0.66** | **67.69** | **0.00** | **28.91** | **0.51** | **74.15** | **0.35** | **53.98** |

Table 1: Performance comparison across datasets and base models. Each dataset reports EM and F1 scores. Best values in each column are **bold**.

## 5.1 EXPERIMENTAL SETUP

**Datasets.** To evaluate the performance of Token-Guard, we conduct experiments across six standard HALU datasets (Ravi et al., 2024): **FinanceBench** (Islam et al., 2023), **DROP** (Dua et al., 2019), **COVID-QA** (Möller et al., 2020), **PubMedQA**) (Jin et al., 2019), **HaluEval** (Li et al., 2023), and **RAGTruth** (Wu et al., 2023). More details are in Appendix E.

**Baselines and Backbone LLMs.** We compare Token-Guard with **Basemodels**, **Chain-of-Thoughts** (Wei et al., 2022), **Tree-of-Thought** (Yao et al., 2023), **Guided Decoding** (Xie et al., 2023) and **Predictive Decoding** (Ma et al., 2024) at two models Meta-Llama-3.1-8B-Instruct (Grattafiori et al., 2024)(used as the default backbone in all subsequent experiments) and Qwen3-8B (Yang et al., 2025).More details are in Appendix F.

All the experiments are implemented on 1 NVIDIA A40 GPU(40GB). The softmax temperature is set to 0.3, the sampling temperature is set to 0.4. More details are in Appendix H.3.

**Evaluation Metrics.** We evaluate Token-Guard and the baselines with three metrics: Exact Match (EM), F1, and BLEU. EM checks exact matches, F1 measures token-level overlap, and BLEU evaluates $n$-gram quality and fluency (Luo et al., 2025c). These metrics together assess factual accuracy, semantic coverage, and generation quality; formal definitions are in Appendix G.

**Hyperparameter Details.** The hyperparameters of Token-Guard follow a practical design; their rationale, functional roles, tuning procedure, and resulting configuration are detailed in Appendix H.

## 5.2 MAIN RESULTS (RQ1)

We compare **Token-Guard** with baselines across different backbone models. As shown in Table 1, **Token-Guard** achieves the highest average EM/F1 across both LLMs:0.29/51.03 on *Meta-Llama-3.1-8B-Instruct* and 0.35/53.98 on *Qwen3-8B*, which demonstrats stable overall performance. .

The relative advantage of Token-Guard varies across datasets due to its adaptive self-checking. On tasks requiring multi-step reasoning and strict factual consistency (e.g., DROP_nfl), token-level and segment-level scoring suppress hallucinations and improve logical correctness, leading to the largest gains. On knowledge-intensive tasks (e.g., **PubMedQA**, **CovidQA**), improvements are smaller, as Token-Guard reduces hallucinations but cannot compensate for missing domain knowledge. Notably, EM scores on RAGTruth and PubMedQA are near zero because these datasets provide long-form, flexible reference answers, so even factually correct outputs rarely match exactly.

Overall, Token-Guard provides robust, cross-task reliability, with improvements depending on the dataset's emphasis on factual verification versus domain knowledge.

## 5.3 ABLATION STUDY (RQ2)

We evaluate the contributions of Token-Guard's core components: prompt initialization (P), token-level scoring (T), segment-level scoring (S), and global iteration (G). Table 2 shows that removing any component degrades performance. Token-level scoring most affects EM and F1, due to its crucial role in guiding early-stage token generation, while prompt initialization and segment-level scoring provide stability. Notably, global iteration boosts BLEU, enhancing linguistic fluency.

| Method Variant | DROP_history | | | RAGTruth | | | Avg. | | |
|---|---|---|---|---|---|---|---|---|---|
| | EM | F1 | BLEU | EM | F1 | BLEU | EM | F1 | BLEU |
| **Full Token-Guard** | 0.48 | 68.52 | 65.21 | 0.02 | 43.94 | 38.27 | 0.25 | 56.23 | 51.74 |
| *w/o Prompt* | 0.32 | 55.23 | 51.48 | 0.01 | 32.50 | 27.92 | 0.17 | 43.87 | 39.70 |
| *w/o Token-Level Scoring* | 0.28 | 47.51 | 44.88 | 0.00 | 27.10 | 25.05 | 0.14 | 37.31 | 34.97 |
| *w/o Segment-Level* Scoring | 0.43 | 60.10 | 59.55 | 0.01 | 39.20 | 33.08 | 0.22 | 49.65 | 46.32 |
| *w/o Global Iteration* | 0.42 | 63.05 | 52.37 | 0.01 | 41.05 | 20.14 | 0.22 | 52.05 | 36.26 |

Table 2: Ablation study of Token-Guard on representative datasets.

## 5.4 ANALYSIS OF TOKEN-GUARD'S EFFECT ON FACTUAL PRECISION (RQ3)

As shown in Figure 4, to evaluate Token-Guard's effect on factual precision, we analyze it from (a) token-level F1 across four datasets and (b) accuracy of estimated step values.

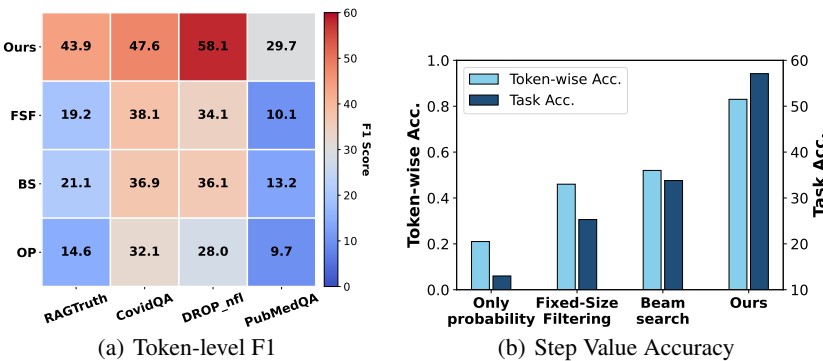

(a) Token-level F1         (b) Step Value Accuracy

Figure 4: Comparison of Token-Guard and token-level decoding methods on factual precision.

**Token-level Precision Advantage.** Token-Guard consistently outperforms other token-level methods, including Fixed-Size Filtering (FSF), Beam Search (BS), and Only Probability (OP)(Wang et al., 2025), demonstrating its effectiveness in retaining factual tokens.

**Reliability Across Generation Steps.** The token-level accuracy is positively correlated with the correctness of the final answer. By ensuring high accuracy at each generation step, Token-Guard improves overall task performance, highlighting the importance of its token-level self-correction mechanism. Please refer to Appendix G for details.

## 5.5 ANALYSIS OF TOKEN-GUARD'S IMPACT ON RELEVANCE (RQ4)

As shown in Figure 5, to evaluate Token-Guard's impact on relevance, we analyze it from (a) segment-level F1 across four datasets and (b) BLEU scores across six datasets.

**Improved Segment-level Precision.** As shown in Figure 5a, Token-Guard substantially outperforms SAS (Self-Adaptive Sampling) and LSTM (Long Short-Term Memory based decoder) (Ali et al., 2024) in segment-level F1 across all datasets, which demonstrates its superior ability to preserve relevant segments and suppress irrelevant fragments during decoding.

**Robust Relevance Across Benchmarks.** As shown in Figure 5b, Token-Guard consistently attains the highest BLEU scores across six datasets, e.g., 63.19 on DROP_history and 75.13 on HaluEval, confirming its robustness in maintaining output relevance and fluency under diverse tasks.

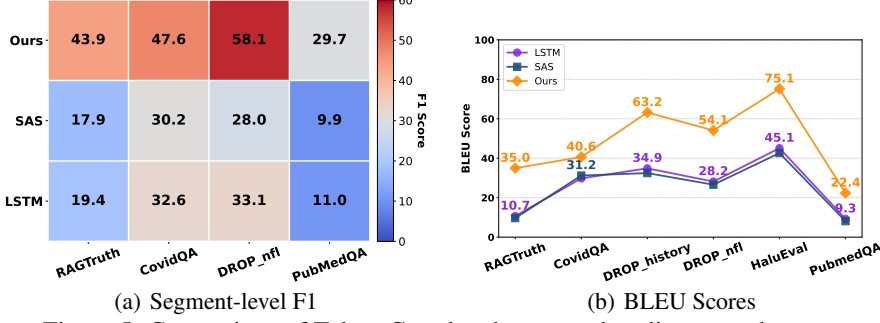

(a) Segment-level F1         (b) BLEU Scores

Figure 5: Comparison of Token-Guard and segment baselines on relevance.

### 5.6 ANALYSIS OF TOKEN-GUARD'S ROLE IN LOGICAL CONSISTENCY AND DYNAMIC RESOURCE MANAGEMENT (RQ5)

As shown in Table 3 and Figure 6, we evaluate Token-Guard from the perspective of its ability to improve generation quality, illustrating how its multi-stage filtering process preserves factual correctness and stabilizes reasoning under ambiguous or underspecified contexts.

**Token-Guard Performance Across Key Dimensions.**

Token-Guard demonstrates clear gains across multiple evaluation dimensions, with notable improvements in Factual Precision, Logical Coherence and Relevance. Its token- and segment-level checks help suppress hallucinations and select more reliable fragments, while iterative refinement maintains logical consistency throughout the reasoning process. The slightly lower score in Conciseness reflects its tendency to provide more detailed explanations, prioritizing accuracy and completeness over brevity. Please refer to Appendix G for details.

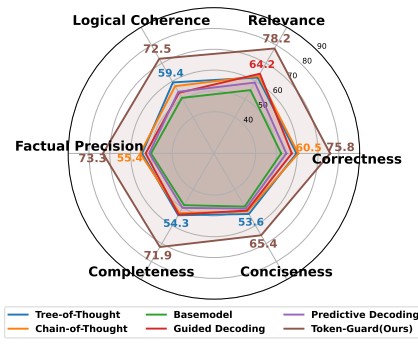

Figure 6: Generation Evaluations.

**Case Study Illustration.** Table 3 presents an example from DROP_nfl: baseline methods either report "insufficient information" or misreport values, while Token-Guard correctly identifies the 61-yard scoring play, preserving both semantic and factual correctness through dynamic local regeneration.Please refer to Appendix A.2 for passage.

| Query | How many yards did gore score the first half ? | | | | |
|---|---|---|---|---|---|
| Golden Answers | [61] | | | | |
| Methods | **Tree-of-Thought** | | **Basemodel** | | **Chain-of-Thoughts** |
| Generation | <reasoning_steps>Step 1: Identify the relevant sentence from the passage "RB Frank Gore set a franchise record with 148 yards rushing in the first half and scored on a 61-yard run before leaving with a concussion."Step 2: Extract the required information. The question asks for the number of yards Gore scored in the first half.Step 3: Determine the answer. The passage states that Gore rushed for 148 yards in the first half and scored on a 61-yard run. However, it does not explicitly specify how many yards he scored in total during the first half.</reasoning_steps> <answer>The passage does not provide enough information to determine the number of yards Gore scored in the first half.</answer> | | <reasoning_steps>The passage does not provide enough information to answer the question about how many yards Gore scored in the first half. It only mentions that Gore set a franchise record with 148 yards rushing in the first half, but it does not mention his scoring yards.</reasoning_steps> <answer>The passage does not provide enough information.</answer> | | <reasoning_steps>Step 1: Identify the relevant sentence from the passage "RB Frank Gore set a franchise record with 148 yards rushing in the first half and scored on a 61-yard run before leaving with a concussion." Step 2: Extract the information required to answer the question. The information required is the number of yards RB Frank Gore scored in the first half. Step 3: Determine the answer based on the extracted information. The passage states that Gore rushed for 148 yards in the first half and scored on a 61-yard run, but it does not explicitly state how many yards he scored in that half.<answer>The passage does not provide enough information to determine the number of yards Gore scored in the first half.</answer> |
| Evaluation Score | F1 | BLEU | F1 | BLEU | F1 | BLEU |
| | 0.00 | 0.00 | 0.00 | 0.00 | 0.00 | 0.00 |
| Methods | **Guided Decoding** | | **Predictive Decoding** | | **Token-Guard(Ours)** |
| Generation | <reasoning_steps>The reasoning steps are: Identify the relevant information in the passage: RB Frank Gore set a franchise record with 148 yards rushing in the first half and scored on a 61-yard run before leaving with a concussion. Extract the specific information about Gore's yards in the first half: The passage states that Gore rushed for 148 yards in the first half. Answer the question: The passage does not mention Gore scoring 148 yards, only that he scored on a 61-yard run. However, it does provide his total rushing yards in the first half as 148. However, based on the passage, the correct answer is:</reasoning_steps> <answer>148 yards.</answer> | | <reasoning_steps>The reasoning steps are: Identify the information in the passage: The passage states that RB Frank Gore set a franchise record with 148 yards rushing in the first half and scored on a 61-yard run before leaving with a concussion. Extract the relevant detail about his first-half performance: The passage explicitly mentions that Gore rushed for 148 yards in the first half. Answer the question: The question asks about the number of yards Gore gained in the first half. The passage specifies 148 rushing yards, not scoring yards.</reasoning_steps> <answer>148.</answer> | | <reasoning_steps>The reasoning steps are: 1.The passage states that RB Frank Gore set a franchise record with 148 yards rushing in the first half and scored on a 61-yard run before leaving with a concussion. 2.The question asks specifically for the number of yards Gore scored in the first half, not his total rushing yards. 3.By examining the passage, the only scoring play mentioned in the first half is the 61-yard run.Therefore, the answer is the yards gained on that scoring play.</reasoning_steps> <answer>61 yards</answer> |
| Evaluation Score | F1 | BLEU | F1 | BLEU | F1 | BLEU |
| | 0.00 | 0.00 | 0.00 | 0.00 | 66.67 | 50.00 |

Table 3: Case study on generation quality under a query from DROP_nfl, passage is in Appendix.

### 5.7 ANALYSIS OF TOKEN-GUARD'S TIME AND MEMORY CONSUMPTION (RQ6)

Here we examine Token-Guard's computational efficiency, analyzing time overhead and memory usage across decoding stages. We focus on two aspects: (a) runtime, including per sample latency and throughput, and (b) memory across token level, segment level, and global level stages. These measurements reveal Token-Guard's resource implications.

**Time and Throughput Analysis.** As shown in Table 4, Token-Guard exhibits distinct time, token, and throughput patterns across datasets. Its dynamic iteration enables efficient computation on **PubMedQA** and **HaluEval**, while allocating more runtime and output tokens for challenging datasets such as **CovidQA** and **RAGTruth**. Notably, despite higher token budgets in some settings,

Token-Guard maintains competitive or superior throughput, reflecting its ability to balance iterative refinement with efficient decoding.

| Method | RAGTruth | CovidQA | PubmedQA | HaluEval | Avg. |
|---|---|---|---|---|---|
| | Time · Token · Tokens/sec | Time · Tokens · Tokens/sec | Time · Tokens · Tokens/sec | Time · Tokens · Tokens/sec | Time · Tokens · Tokens/sec |
| Chain-of-Thoughts | 11 · 899 · 80.71 | 9 · 670 · 73.14 | 11 · 998 · 88.45 | 7 · 543 · 77.33 | 9.65 · 777.08 · 79.41 |
| Tree-of-Thought | 57 · 3687 · 64.49 | 67 · 3423 · 51.09 | 58 · 4253 · 73.38 | 35 · 2786 · 79.49 | 54.29 · 3537.19 · 67.11 |
| Predictive Decoding | 107 · 12058 · 112.82 | 138 · 6473 · 47.07 | 79 · 8416 · 106.06 | 58 · 5954 · 102.20 | 95.50 · 8225.61 · 92.54 |
| Guided Decoding | 110 · 17234 · 155.96 | 189 · 13525 · 71.64 | 103 · 18733 · 182.05 | 77 · 11689 · 151.15 | 119.38 · 15295.41 · 140.20 |
| Token-Guard | 69 · 18024 · 262.81 | 301 · 17474 · 58.10 | 32 · 7699 · 240.58 | 54 · 5254 · 97.54 | 113.29 · 12112.95 · 164.76 |

Table 4: Time, output token consumption, and normalized throughput (tokens/sec).

**Memory Consumption Analysis.** On the **HaluEval** dataset, Token-Guard shows a multi-stage memory profile: token-level states are buffered efficiently (1.2–9.4%), segment-level briefly peaks (21.6–19.6%), and global-level stabilizes around 10.3–19.0%. Overall, peak (21.6%) and average ( 16.3%) memory remain competitive with baselines. Table 5 summarizes peak memory, average memory and runtime for Token-Guard and other methods.

| Method | Peak Memory (%) | Avg Memory (%) | Runtime (s) |
|---|---|---|---|
| Guided Decoding | 21.4 | 15.0 | 77.31 |
| Chain-of-Thoughts | 21.3 | 14.5 | 7.02 |
| Tree-of-Thought | 21.4 | 17.8 | 35.04 |
| Predictive Decoding | 4.1 | 3.7 | 58.26 |
| Token-Guard | 21.6 | 16.3 | 53.89 |

Table 5: Memory and runtime comparison for Token-Guard and baseline methods.

## 5.8 ANALYSIS OF TOKEN-GUARD'S PORTABILITY ACROSS BACKBONE MODELS (RQ7)

Table 6 reports the performance of Token-Guard and several baseline decoding strategies, evaluated on both **Llama-3.2-3B-Instruct** and **Llama-3.2-13B-Instruct** backbone models. Predictive Decoding (PD) is omitted on the 13B backbone because it triggered out-of-memory (OOM) errors during decoding.

Token-Guard achieves the highest F1 and BLEU on **DROP_history** and **HaluEval**, showing consistent gains on both model scales. These improvements indicate that Token-Guard scales favorably with larger backbone capacity.

On **RAGTruth**, Token-Guard performs lower

Table 6: Performance on 3B and 13B models.

| Method | RAGTruth | | DROP_history | | HaluEval | |
|---|---|---|---|---|---|---|
| | F1 | BLEU | F1 | BLEU | F1 | BLEU |
| *3B Model* | | | | | | |
| BaseModel | 12.11 | 6.29 | 42.21 | 39.31 | 42.95 | 39.95 |
| Chain-of-Thoughts | **32.70** | **25.48** | 31.60 | 26.04 | 50.88 | 48.44 |
| Tree-of-Thought | *27.05* | *21.16* | *39.84* | *36.14* | 59.22 | 55.11 |
| Predictive Decoding | 21.72 | 14.59 | 34.19 | 31.07 | 57.55 | 53.97 |
| Guided Decoding | 30.50 | 18.70 | 32.60 | 28.83 | *63.47* | *59.11* |
| **Token-Guard(Ours)** | 15.66 | 13.45 | **43.09** | **40.57** | **68.21** | **63.83** |
| *13B Model* | | | | | | |
| BaseModel | 22.47 | 15.26 | 46.32 | 42.74 | 35.30 | 31.40 |
| Chain-of-Thoughts | 32.25 | 27.67 | 33.90 | 30.06 | 49.43 | 44.01 |
| Tree-of-Thought | 32.93 | 23.70 | 43.97 | 38.25 | 62.20 | 58.07 |
| Predictive Decoding | – | – | – | – | – | – |
| Guided Decoding | 29.34 | 22.01 | 40.41 | 36.71 | 63.42 | 58.72 |
| **Token-Guard(Ours)** | **33.14** | **30.41** | **51.17** | **47.64** | **72.66** | **68.81** |

on the 3B backbone due to the dataset's long-context structure, which is challenging for smaller models. The 13B model alleviates this issue to some extent, offering more stable decoding. Overall, Token-Guard remains effective across backbone sizes, with limitations mainly arising from extremely large-context settings. The results confirm that Token-Guard is portable across model scales without requiring additional tuning.

## 6 CONCLUSION

In this work, we introduce Token-Guard, a token-level hallucination-controlled decoding framework for large language models. By embedding self-checking at each reasoning step and leveraging a three-stage filtering pipeline, Token-Guard detects and suppresses hallucinated tokens, ensuring multi-step reasoning preserves factual and logical accuracy. Experiments on HALU benchmarks show that Token-Guard substantially improves generation quality, achieving higher F1 scores and stronger alignment with ground-truth responses compared to conventional decoding methods. Overall, Token-Guard maintains efficient resource usage, demonstrates portability across model scales, offers a reliable approach for mitigating hallucinations in large language model generation.

ACKNOWLEDGMENTS

This work is supported by the National Key Research and Development Program of China under Grant 2024YFC3308500, Beijing Municipal Natural Science Foundation under Grant L251042, National Natural Science Foundation of China under Grant 62406036, China Postdoctoral Science Foundation under Grant 2025M781457, and also sponsored by the State Key Laboratory of Networking and Switching Technology under Grant NST20250110.

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

APPENDIX

## A  INPUTS USED IN TOKEN-GUARD

### A.1  TOKEN-GUARD PROMPT

As shown in Figure 8, we detail the dataset-specific prompts used in our experiments.

In **PubMedQA**, answers must start with "Yes./No./Maybe." and provide a concise one-sentence conclusion, preserving key medical terms and conditions. In **FinanceBench**, outputs must exactly match the ground-truth format (USD amounts, percentages, ratios) and only use provided financial data without intermediate calculations. In **History**, answers must use only numbers and entities explicitly mentioned in the passage, with insufficient information handled by a fixed rejection statement. In retrieval-augmented QA like **RagTruth**, answers are limited to the given passages, including all factual details, or else return a standardized refusal.

Combining domain-specific constraints with a fallback general prompt improves answer consistency and reduces hallucination risk across domains.

```
prompt_map = {
  'pubmedqa': (
    "You will be given a PubMed-style passage and a Yes/No/Maybe question.\n"
    "Answer rules:\n"
    "1. Begin with exactly one of: \"Yes.\" / \"No.\" / \"Maybe.\"\n"
    "2. Summarize the main conclusion from the passage in exactly ONE short sentence (≤25 words).\n"
    "3. Preserve key phrases and medical terms from the passage; do not replace them with synonyms.\n"
    "4. Always include explicitly stated conditions, subgroups, or limitations if they appear in the conclusion.\n"
    "5. Do NOT add recommendations, explanations, or new information.\n"
    "Answer:[Yes./No./Maybe. + short sentence]"
  ),
  'financebench': (
    "You are an equity research analyst. Answer the question using **only the data provided**.\n"
    "1. Always produce a single-line final answer.\n"
    "2. Do not show calculations, reasoning, or commentary.\n"
    "3. Match the exact format of the ground truth (e.g., \"$360000.00\" for USD thousands, \"$7223.00\" for USD millions, \"$4.90\" for USD billions, \"34.7%\" for percentages, \"1.08\" for ratios).\n"
    "4. If the answer is not directly available from the statements, output: \"Unable to answer based on given data.\"\n"
    "Example: Q: How much was Boeing's FY2017 interest expense (USD thousands)?\n"
    "A: Answer: $360000.00\n"
    "At the end, output: Answer:[your answer here]."
  ),
  'history': (
    "You will be presented with a question.\n"
    "1. Question Analysis: Determine the expected answer type (e.g., 'how many' → number, 'what year' → year, 'what/which/who' → span, 'when' → time).\n"
    "2. Number Handling: Only use numbers explicitly from the passage; do not assume or invent; handle time and organizational changes carefully.\n"
    "3. Information Extraction: Extract exact names, numbers, and events exactly as stated in the passage.\n"
    "4. Insufficient Information: If the passage does not provide enough information, answer with 'The passage does not provide enough information to answer this question.'\n"
    "5. Multiple Choice: If the question contains 'or', answer strictly using the given options.\n"
    "At the end, output: Answer:[your answer here]."
  ),
  'ragtruth': (
    "You are given passages and a question. Follow these steps:\n"
    "Answer the question using only the information from the given passages.\n"
    " - Include specific examples, numbers, or comparisons if mentioned.\n"
    " - Include all details that support the answer.\n"
    " - Do not add external information.\n"
    " - If the passages do not contain sufficient information, answer: \"Unable to answer based on given passages.\"\n"
    "At the end, output: Answer:[your answer here]"
  ),
}
default_prompt = (
  "You will be presented with a question.\n"
  "Answer the user's question strictly based on the given information.\n"
  "Do not make up information.\n"
  "At the end, output: Answer:[your answer here]."
)
```

Figure 7: Prompt for Token-Guard.

### A.2  CASE STUDY PASSAGE

San Francisco kept Detroit in the game with missed opportunities, then made just enough plays for a rare two-game winning streak and its first road win. RB Frank Gore set a franchise record with 148 yards rushing in the first half and scored on a 61-yard run before leaving with a concussion. San Francisco led 13-3 at halftime after scoring on three of its first four drives, wishing it had a bigger lead after out gaining Detroit 247-102 yards and recovering a fumble without giving up a turnover. The 49ers had chances to go ahead big in the third quarter because Detroit had two turnovers on its first three plays, but they came away with only a field goal and a 13-point lead. Gore then caught a 7-yard pass, wobbled off the field and didn't return. QB Alex Smith's fumble midway through the third quarter set up a score that helped the Lions get back in the game. The 49ers drove down the field to set up K Joe Nedney's fourth field goal, a key score because it made Detroit go for a TD instead of kicking a tying field goal late in the game. S Keith Lewis intercepted QB Jon Kitna's pass at the 49ers' 2 with 2½ minutes left and San Francisco now 4-5 picked up the one first down it needed to seal the game, winning consecutive games for the second time since 2003. RB Frank Gore finished with career-high 159 yards rushing and San Francisco's QB Alex Smith was 14 of 20 for 136 yards with a fumble. WR Arnaz Battle caught six passes for 55 yards, and converted a third-and-4 on the final drive to allow the 49ers to run out the clock. The 49ers' defense again came up huge, allowing only 273 total offensive yards and forcing 4 turnovers. LB Brandon Moore was yet again the story, leading the team with 9 tackles, 2 sacks and forcing 2 turnovers. With the win, the 49ers advanced to 4-5.

Figure 8: Prompt for Token-Guard.

# B THEORETICAL PROOF

## B.1 PROOF OF PROPOSITION 1

**Proposition 1.** *Token-level self-checking reduces expected hallucination of generated sequences.*

*Proof.* Let $\mathbf{a}_t$ denote the token generated at step $t$, with model probability $p_\theta(\mathbf{a}_t \mid \mathbf{s}_t)$, where $\mathbf{s}_t$ is the current state (including context and previously generated tokens). Each token is associated with a hallucination score $F_{\mathrm{halu}}(\mathbf{a}_t) \in [0, 1]$, measuring its likelihood of being factually incorrect.

Token-Guard modifies the generation probability to penalize hallucinations:

$$p_{\mathrm{guard}}(\mathbf{a}_t \mid \mathbf{s}_t) = \frac{p_\theta(\mathbf{a}_t \mid \mathbf{s}_t) \exp\left(-\gamma F_{\mathrm{halu}}(\mathbf{a}_t)\right)}{Z_t},$$
$$Z_t = \sum_{\mathbf{a} \in \mathcal{V}} p_\theta(\mathbf{a} \mid \mathbf{s}_t) \exp\left(-\gamma F_{\mathrm{halu}}(\mathbf{a})\right), \tag{19}$$

where $\gamma > 0$ controls hallucination penalization strength, $Z_t$ ensures normalization, and $\mathcal{V}$ denotes the set of all possible tokens in the vocabulary.

Define the expected hallucination at step $t$:

$$\mathbb{E}_{p_{\mathrm{guard}}}[F_{\mathrm{halu}}] = \sum_{\mathbf{a}_t} p_{\mathrm{guard}}(\mathbf{a}_t \mid \mathbf{s}_t) F_{\mathrm{halu}}(\mathbf{a}_t). \tag{20}$$

**Information-theoretic view:** Define the conditional entropy of the token under the original model and Token-Guard:

$$H_\theta(\mathbf{A}_t \mid \mathbf{s}_t) = -\sum_{\mathbf{a}_t} p_\theta(\mathbf{a}_t \mid \mathbf{s}_t) \log p_\theta(\mathbf{a}_t \mid \mathbf{s}_t),$$
$$H_{\mathrm{guard}}(\mathbf{A}_t \mid \mathbf{s}_t) = -\sum_{\mathbf{a}_t} p_{\mathrm{guard}}(\mathbf{a}_t \mid \mathbf{s}_t) \log p_{\mathrm{guard}}(\mathbf{a}_t \mid \mathbf{s}_t). \tag{21}$$

Here, "Facts" denotes the factual information or evidence relevant to the generation task. The conditional mutual information between the generated token and factual evidence is

$$I(\mathbf{A}_t; \mathrm{Facts} \mid \mathbf{s}_t) = H(\mathbf{A}_t \mid \mathbf{s}_t) - H(\mathbf{A}_t \mid \mathrm{Facts}, \mathbf{s}_t). \tag{22}$$

Since Token-Guard down-weights high-hallucination tokens, we have

$$\mathbb{E}_{p_{\mathrm{guard}}}[F_{\mathrm{halu}}] \leq \mathbb{E}_{p_\theta}[F_{\mathrm{halu}}],$$
$$I(\mathbf{A}_t; \mathrm{Facts} \mid \mathbf{s}_t)_{\mathrm{guard}} \geq I(\mathbf{A}_t; \mathrm{Facts} \mid \mathbf{s}_t)_\theta, \tag{23}$$

i.e., expected hallucination decreases and mutual information with factual evidence increases.

**Summary:** Token-level self-checking both reduces expected hallucination per token and increases factual alignment, providing a theoretically grounded basis for subsequent segment-level and trajectory-level selection. This effectively improves the model's factual precision during generation. □

## B.2 PROOF OF PROPOSITION 2

**Proposition 2.** *Segment-level scoring with local refinement improves generation quality and reduces hallucination.*

*Proof.* Let a candidate segment $C_k$ consist of tokens $\{\mathbf{a}_{t_1}, \ldots, \mathbf{a}_{t_n}\}$, each with hallucination score $F_{\mathrm{halu}}(\mathbf{a}_{t_i})$ and weight $w_i$ ($\sum_i w_i = 1$). Define the segment-level score:

$$F_{\mathrm{seg}}(C_k) = \sum_{i=1}^{n} w_i F_{\mathrm{halu}}(\mathbf{a}_{t_i}) + \beta H_k, \tag{24}$$

where $H_k$ measures structural or semantic consistency, and $\beta > 0$ balances hallucination vs. structure.

**Expected segment-level hallucination:**

$$\mathbb{E}[F_{\text{seg}}] = \sum_{i=1}^{n} w_i \mathbb{E}[F_{\text{halu}}(\mathbf{a}_{t_i})] + \beta \mathbb{E}[H_k]. \tag{25}$$

When $F_{\text{seg}}(C_k)$ falls below a threshold $\tau_{\text{seg}}$, a local refinement window $W^{(l)} = \{\mathbf{a}_{i-1}, \mathbf{a}_i^{\text{low}}, \mathbf{a}_{i+1}\}$ is extracted, where $\mathbf{a}_i^{\text{low}}$ is the lowest-scoring token. The refined window is generated by

$$W^{(l)'} = \text{LLM\_refine}(W^{(l)} \mid C_k \setminus W^{(l)}), \tag{26}$$

yielding an updated segment $C_k' = C_k \setminus W^{(l)} \cup W^{(l)'}$.

**Information-theoretic view:** Let the mutual information between a segment and the factual source be

$$I(C_k; \text{Facts}) = \sum_{i=1}^{n} w_i I(\mathbf{a}_{t_i}; \text{Facts}) + I(H_k; \text{Facts}). \tag{27}$$

Refinement improves the weakest sub-window, so that

$$\mathbb{E}[I(C_k'; \text{Facts})] \geq \mathbb{E}[I(C_k; \text{Facts})], \tag{28}$$

ensuring that corrected segments are more consistent with factual knowledge.

**Summary:** Segment-level scoring combined with local refinement guarantees that unreliable subparts are iteratively improved. This reduces expected hallucination while enhancing semantic coherence, factual alignment, and generation relevance, thereby improving the overall generation quality. □

### B.3 PROOF OF PROPOSITION 3

**Proposition 3.** *Iterative refinement over candidate sequences reduces hallucination and improves final response reliability by enhancing logical consistency.*

*Proof.* Let $\{\tau_i\}_{i=1}^N$ be candidate trajectories, each with segments $\{C_1^{(i)}, \ldots, C_K^{(i)}\}$. Define the trajectory-level score:

$$F_{\text{final}}(\tau_i) = \lambda \sum_{k=1}^{K} F_{\text{seg}}(C_k^{(i)}) + (1-\lambda) \sum_{t=1}^{T} F_{\text{halu}}(\mathbf{a}_t^{(i)}), \quad \lambda \in [0, 1]. \tag{29}$$

Select the optimal trajectory hierarchically via clustering:

$$\tau_j^* = \arg \min_{\tau_i \in \mathcal{C}_j} F_{\text{final}}(\tau_i), \tag{30}$$

$$\tau_{\text{final}} = \arg \min_{j} F_{\text{final}}(\tau_j^*). \tag{31}$$

**Iterative refinement for global consistency.** To further enhance logical coherence, we introduce a global iterative update:

$$F^{(t+1)}(\tau_i) = \alpha \, F_{\text{final}}^{(t)}(\tau_i) + (1-\alpha) \frac{1}{N-1} \sum_{j \neq i} \text{Sim}(\tau_i, \tau_j), \tag{32}$$

where $\text{Sim}(\tau_i, \tau_j)$ measures logical agreement between trajectories and $\alpha \in (0, 1)$ balances local reliability and global consensus. After convergence, the final trajectory is

$$\tau_{\text{final}}^* = \arg \min_{\tau_i \in \mathcal{R}} F^{(T)}(\tau_i), \tag{33}$$

which incorporates both hallucination minimization and global logical consistency.

**Information-theoretic view:** The mutual information between trajectory and factual knowledge is

$$I(\tau_i; \text{Facts}) = \sum_{k=1}^{K} I(C_k^{(i)}; \text{Facts}) + \sum_{t=1}^{T} I(\mathbf{a}_t^{(i)}; \text{Facts}). \tag{34}$$

Iterative refinement maximizes $I(\tau_i; \text{Facts})$ under consensus constraints, ensuring that the selected trajectory carries maximal factual content and consistent logical reasoning.

**Summary:** Global multi-stage selection with iterative refinement reduces hallucinations, enforces logical consistency, and dynamically corrects low-confidence fragments while controlling computational resources.

$\square$

## C  TOKEN-GUARD ALGORITHM DETAILS

To illustrate the mechanism of Token-Guard, we present its full workflow in Algorithm 1, comprising three key stages designed to mitigate hallucinations through iterative self-checking and global consistency evaluation.

---

**Algorithm 1** Token-Guard: Multi-Stage Hallucination-Controlled Generation

---

**Require:** Input $x$, language model LLM, thresholds $\{\tau_{\text{token}}, \tau_{\text{seg}}^{\text{low}}, \tau_{\text{seg}}^{\text{high}}, \tau_{\text{global}}\}$, max steps $N_{\max}$
**Ensure:** Final hallucination-controlled answer $y$
1: **// Phase 1: Token-Level Generation & Verification**
2: Initialize latent environment $s_1 \leftarrow \emptyset$, reasoning chain $R \leftarrow \emptyset$
3: **for** $t = 1$ **to** $T$ **do**
4:     Sample candidates $\{a_t^{(i)}\}$ and hidden states $\{h_t^{(i)}\}$ from LLM
5:     Compute score $F_{\text{halu}}^{\text{token}}(a_t^{(i)} \mid s_t)$ via semantic similarity and probability
6:     Select $a_t^*$ satisfying $\tau_{\text{token}}$; $s_{t+1} \leftarrow s_t \cup \{h_t^{(a_t^*)}\}$; Append $a_t^*$ to current segment $C_k$
7: **end for**
8: **// Phase 2: Segment-Level Evaluation and Local Refinement**
9: **for** each completed segment $C_k = \{a_{t_1}, \ldots, a_{t_n}\}$ **do**
10:     Compute $H_k = \sum w_i h_{t_i}^{(i)}$ and segment score $F_{\text{halu}}^{\text{seg}}(C_k)$
11:     **if** $F_{\text{halu}}^{\text{seg}}(C_k) \geq \tau_{\text{seg}}^{\text{high}}$ **then** Add $C_k$ to valid pool and append to $R$
12:     **else if** $\tau_{\text{seg}}^{\text{low}} \leq F_{\text{halu}}^{\text{seg}}(C_k) < \tau_{\text{seg}}^{\text{high}}$ **then**
13:         **while** $F_{\text{halu}}^{\text{seg}}(C_k) < \tau_{\text{seg}}^{\text{high}}$ **and** $r < N_{\max}$ **do**
14:             $a_{\text{low}} \leftarrow \arg\min_i F_{\text{halu}}^{\text{token}}$; $W_k^{(l)'} \leftarrow \text{LLM\_refine}(W_k^{(l)} \mid \text{context}, H_k)$
15:             Update $C_k \leftarrow (C_k \setminus W_k^{(l)}) \cup W_k^{(l)'}$; Recompute $H_k$ and $F_{\text{halu}}^{\text{seg}}(C_k)$; $r \leftarrow r + 1$
16:         **end while**
17:         **if** $F_{\text{halu}}^{\text{seg}} \geq \tau_{\text{seg}}^{\text{high}}$ **then** Accept $C_k$; append to $R$ **else** Discard $C_k$
18:     **else** Discard $C_k$
19:     **end if**
20: **end for**
21: **// Phase 3: Global Re-thinking and Feedback Correction**
22: Initialize iteration $i \leftarrow 1$
23: **while** $i \leq N_{\max}$ **do**
24:     Compute $F_{\text{fact}}(R)$ and Logical Coherence $F_{\text{logic}}(R)$ via segment representations $\{H_k\}$
25:     Compute Global Score $F_{\text{global}}(R) = \frac{F_{\text{fact}} \cdot F_{\text{logic}}}{F_{\text{fact}} + F_{\text{logic}} - F_{\text{fact}} \cdot F_{\text{logic}}}$
26:     **if** $F_{\text{global}}(R) \geq \tau_{\text{global}}$ **then return** answer $y = R$
27:     **else** Adjust $\tau_{\text{seg}}^{\text{adj}} \leftarrow f(F_{\text{fact}}, F_{\text{logic}})$; Refine chain $R$ via Stage 2 logic; $i \leftarrow i + 1$
28:     **end if**
29: **end while**
30: **return** "cannot answer"                    $\triangleright$ Exceeded $N_{\max}$ global iterations

---

## D  COMPREHENSIVE EVALUATION

To evaluate the potential of TOKEN-GUARD across a variety of datasets and tasks, we conduct experiments on seven QA and reasoning benchmarks as well as the open-domain StrategyQA dataset. For StrategyQA (Geva et al., 2021), we use the LLaMA-7B model and compare against the Decoding by Contrasting Layers method **DoLa** (Chuang et al., 2024), following the accuracy-based evaluation protocol reported in the original DoLa paper.

| Method | Accuracy (%) | Base Acc. (%) | Improved (%) | Improvement Rate (%) |
|---|---|---|---|---|
| DoLa | 64.1 | 60.1 | 4.0 | 6.66 |
| Token-Guard | 69.4 | 62.3 | 7.1 | 11.40 |

Table 7: Results on the STRATEGYQA dataset using LLaMA-7B.

From Table 7, we observe that TOKEN-GUARD achieves 69.4% accuracy, outperforming DoLa by 5.3 points and nearly doubling the improvement rate (11.40% vs. 6.66%). This demonstrates that explicit token-level hallucination scoring yields benefits even in open-domain settings where external grounding is limited. The method forms more reliable reasoning trajectories and reduces errors caused by locally inconsistent token transitions.

## D.1 MULTI-DATASET BENCHMARK RESULTS

To better assess the robustness of Token-Guard , we further incorporate several widely used refinement-based baselines,including SELF-REFINE(Madaan et al., 2023), Self-RAG(Asai et al., 2024), and Self-Reflection(Ji et al., 2023a), covering both retrieval-augmented and self-improvement paradigms. These baselines allow a comprehensive comparison across different families of hallucination-mitigation strategies.

| Method | FinanceBench | RAGTruth | CovidQA | DROP-h | DROP-n | PubmedQA | Halueval | Avg |
|---|---|---|---|---|---|---|---|---|
| BaseModel | 16.00 | 14.71 | 32.33 | 44.21 | 39.10 | 9.55 | 42.16 | 28.29 |
| Self-RAG | 42.74 | 32.91 | 42.48 | 50.13 | 45.29 | 43.21 | 86.70 | 51.78 |
| SELF-REFINE | 17.09 | 28.93 | 38.66 | 42.64 | 47.25 | 19.47 | 71.75 | 37.97 |
| Self-Reflection | 19.37 | 14.82 | 33.76 | 47.48 | 42.05 | 12.83 | 46.21 | 30.93 |
| **Token-Guard (Ours)** | **30.80** | **43.94** | **47.64** | **68.52** | **58.10** | **29.67** | **78.54** | **51.03** |

Table 8: F1 results across seven QA datasets.

Across the seven datasets, we observe clear differences among the baselines. SELF-REFINE and Self-RAG exhibit notable gains on short-text or low-complexity tasks, but their improvements diminish substantially on long-context, high-compositionality datasets such as DROP and CovidQA. Self-RAG in particular is highly dependent on the reliability of its constructed evidence corpus, leading to sharp drops in performance when retrieval becomes unstable. Self-Reflection also shows limited effectiveness, as repeated rounds of regeneration often fail to meaningfully reduce the discrepancy between the predicted and gold answers. In contrast, Token-Guard delivers consistent and robust improvements across all datasets without requiring retrieval or external corpus construction, resulting in strong absolute F1 gains even on the most challenging benchmarks.

## E DATASET DETAILS

We conduct experiments on six widely-used In-Context HALU benchmarks selected from HaluBench (Ravi et al., 2024), covering specialized questions across multiple domains where large models are prone to hallucinate in their generated answers.

- **FinanceBench** (Islam et al., 2023): A dataset of 10,000 financial document QA pairs including tables and bullet points, designed to mimic real-world questions asked by financial analysts.

- **DROP** (Dua et al., 2019): An English reading comprehension benchmark evaluating reasoning ability over passages; we use its `history` and `nfl` subsets for experiments.

- **CovidQA** (Möller et al., 2020): A dataset of 2,000 QA pairs annotated by biomedical experts on COVID-19 related scientific articles.

- **PubMedQA** (Jin et al., 2019): A biomedical QA dataset collected from PubMed abstracts, requiring Yes/No/Maybe answers to research questions, with long-form evidence-based answers.

- **HaluEval** (Li et al., 2023): A hallucination evaluation dataset containing user queries and Chat-GPT responses, with task-specific examples from QA, knowledge-grounded dialogue, and summarization (we use the `qa_samples` subset).

- **RAGTruth** (Wu et al., 2023): A corpus with word-level hallucination annotations for LLM-generated text.

## F    BASELINE DETAILS

Our experiments compare **Token-Guard** with four representative reasoning-based baselines:

- **Auto-Regressive (CoT)**:(Wei et al., 2022) Produces chain-of-thought reasoning through auto-regressive language generation, but is prone to hallucinations due to the lack of self-correction.
- **Tree-of-Thought (ToT)** (Yao et al., 2023): Builds a tree structure for a given problem, where each node represents a reasoning step. We adopt the BFS implementation, which improves exploration and reduces local hallucinations.
- **Guided Decoding** (Xie et al., 2023): Utilizes self-evaluation at each step to perform stochastic beam search, effectively filtering low-quality reasoning paths and mitigating token-level hallucinations.
- **Predictive Decoding** (Ma et al., 2024): Proposes a look-ahead strategy leveraging Model Predictive Control to reweigh LLM distributions, enabling non-myopic language modeling and reducing long-range hallucinations.

## G    EVALUATION DETAILS

We evaluate the performance of Token-Guard using four widely adopted metrics:

**(i) Exact Match (EM).** EM measures whether the predicted answer exactly matches the ground truth. Let $\text{norm}(\cdot)$ denote the normalization function:

$$\text{EM} = \frac{1}{N} \sum_{i=1}^{N} \mathbb{I}\left\{\text{norm}(y_i) = \text{norm}(y_i^\star)\right\}. \quad (35)$$

**(ii) F1 Score.** The F1 score measures the token-level overlap between the predicted answer $y_i$ and the ground-truth answer $y_i^\star$ using the harmonic mean of precision and recall:

$$\text{F1} = \frac{1}{N} \sum_{i=1}^{N} \frac{2 \cdot |\text{tokens}(y_i) \cap \text{tokens}(y_i^\star)|}{|\text{tokens}(y_i)| + |\text{tokens}(y_i^\star)|}. \quad (36)$$

**(iii) BLEU.** BLEU evaluates the n-gram precision of the predicted answer with respect to the ground truth, with a brevity penalty (BP) to penalize overly short outputs. For $n$-gram order up to $K$:

$$\text{BLEU} = \text{BP} \cdot \exp\left(\sum_{n=1}^{K} w_n \log p_n\right), \quad (37)$$

where $p_n$ is the modified $n$-gram precision, $w_n$ is the weight (uniformly $1/K$), and $\text{BP} = \min\left(1, \exp\left(1 - \frac{|y^\star|}{|y|}\right)\right)$.

**(iv) Token-Level Accuracy Across Generation Steps.** To evaluate the reliability of Token-Guard at each generation step, we compute the step-wise correctness:

$$\text{Acc}_t = \mathbf{1}(a_t = a_t^\star), \quad (38)$$

where $a_t$ is the predicted token and $a_t^\star$ is the ground-truth token at step $t$, and $\mathbf{1}(\cdot)$ is the indicator function.

The overall token-level accuracy for a sequence of length $T$ is then

$$\text{Token-level Accuracy} = \frac{1}{T} \sum_{t=1}^{T} \text{Acc}_t. \quad (39)$$

## H    PARAMETER OPTIMIZATION DETAILS

### H.1    CONFIGURATION OF CORE PROCESS PARAMETERS

As shown in Table 9, we summarize the effect of different Token-Guard parameters. Increasing $\lambda$ and $\Delta\tau$ generally improves F1 but also increases time per sample (TPS).

Each Token-Guard hyperparameter has a clear functional role: $\lambda$ and $(\alpha, \beta, \gamma)$ control the interaction between token-level confidence and segment-level semantic consistency, while $\Delta\tau$, $N_{\max}$, and

$M_{\max}$ define the scope and iteration budget for local refinement. A grid search was performed across the full training corpus of all benchmark datasets to select a single, robust configuration balancing F1 and TPS.

| ID | $\lambda$ | $\Delta\tau$ | $(\alpha, \beta, \gamma)$ | $N_{\max}$ | $M_{\max}$ | F1 | TPS (s) |
|----|------|------|-----------------|-----|-----|-------|--------|
| 1 | 0.40 | 0.05 | (0.3,0.3,0.4) | 2 | 2 | 45.32 | 95.21 |
| 2 | 0.40 | 0.10 | (0.4,0.3,0.3) | 2 | 3 | 48.76 | 142.35 |
| 3 | 0.40 | 0.15 | (0.5,0.25,0.25) | 2 | 4 | 50.21 | 187.64 |
| 4 | 0.50 | 0.05 | (0.3,0.3,0.4) | 3 | 2 | 49.12 | 110.87 |
| 5 | 0.50 | 0.10 | (0.4,0.3,0.3) | 3 | 3 | 52.45 | 156.78 |
| 6 | 0.50 | 0.15 | (0.5,0.25,0.25) | 3 | 4 | 55.03 | 201.95 |
| 7 | 0.60 | 0.05 | (0.3,0.3,0.4) | 3 | 2 | 50.12 | 122.56 |
| 8 | 0.60 | 0.10 | (0.5,0.3,0.2) | 3 | 2 | 51.03 | 113.29 |
| 9 | 0.60 | 0.15 | (0.5,0.25,0.25) | 2 | 4 | 53.87 | 180.12 |
| 10 | 0.70 | 0.05 | (0.3,0.3,0.4) | 3 | 2 | 51.56 | 131.45 |
| 11 | 0.70 | 0.10 | (0.4,0.3,0.3) | 3 | 3 | 54.12 | 165.88 |
| 12 | 0.70 | 0.15 | (0.5,0.25,0.25) | 3 | 4 | 56.78 | 198.45 |
| 13 | 0.80 | 0.05 | (0.3,0.3,0.4) | 2 | 2 | 48.92 | 104.36 |
| 14 | 0.80 | 0.10 | (0.4,0.3,0.3) | 2 | 3 | 50.67 | 148.72 |
| 15 | 0.80 | 0.15 | (0.5,0.25,0.25) | 3 | 4 | 57.34 | 205.89 |

Table 9: Grid search results for Token-Guard. Highlighted row indicates the selected high-performance, cost-efficient configuration.

Increasing $N_{\max}$ and $M_{\max}$ leads to a rapid, multiplicative growth in TPS, especially for complex datasets, illustrating the trade-off between accuracy and efficiency. Our chosen configuration (ID 8) with $\lambda = 0.60$, $\Delta\tau = 0.10$, $(\alpha, \beta, \gamma) = (0.5, 0.3, 0.2)$, $N_{\max} = 3$, and $M_{\max} = 2$ achieves F1=51.03 at TPS=113.29, demonstrating high cost-effectiveness.

## H.2 THRESHOLD PROPAGATION FROM TOKEN TO SEGMENT AND GLOBAL LEVELS

Token-Guard thresholds are interdependent. We systematically propagate token-level thresholds $\tau_{\text{token}}$ to segment- and global-level thresholds, ensuring semantic coherence and factual reliability.

**Segment-Level Thresholds** For a candidate segment $C_k$ with $n$ tokens that passed token-level checking, let

$$\bar{F}_{\text{token}}^{\text{seg}} = \frac{1}{n} \sum_{i=1}^{n} F_{\text{halu}}^{\text{token}}(a_{t_i} \mid s_{t_i}) \tag{40}$$

which denotes the average token-level hallucination score. Segment acceptance thresholds are defined as:

$$\tau_{\text{seg}}^{\text{high}} = \bar{F}_{\text{token}}^{\text{seg}} + k_1(C_{\text{seg}} - 0.5), \tag{41}$$

$$\tau_{\text{seg}}^{\text{low}} = \bar{F}_{\text{token}}^{\text{seg}} - k_2(1 - C_{\text{seg}}), \tag{42}$$

where $k_1, k_2 \in [0.1, 0.2]$ control the threshold margin. High-threshold segments are accepted immediately, while moderate ones can undergo local refinement.

**Global Threshold** The global threshold is based on segment-level thresholds and expected consistency:

$$\tau_{\text{global}} = \max\left(\tau_{\text{seg}}^{\text{high}} - \Delta_1, F_{\text{fact}}^{\text{expected}}\right), \tag{43}$$

where $\Delta_1 \in [0.05, 0.1]$ and $F_{\text{fact}}^{\text{expected}} \approx 0.7$.

**Propagation Algorithm**

1. Choose token-level threshold $\tau_{\text{token}}$.

2. Assume segment consistency $C_{\text{seg}} \approx 0.7$.

3. Compute segment average score $\bar{F}_{\text{token}}^{\text{seg}} \approx \tau_{\text{token}}$.

4. Compute segment-level thresholds:

$$\tau_{\text{seg}}^{\text{high}} = \bar{F}_{\text{token}}^{\text{seg}} + k_1(C_{\text{seg}} - 0.5), \quad \tau_{\text{seg}}^{\text{low}} = \bar{F}_{\text{token}}^{\text{seg}} - k_2(1 - C_{\text{seg}})$$

5. Compute global threshold:

$$\tau_{\text{global}} = \max(\tau_{\text{seg}}^{\text{high}} - \Delta_1, F_{\text{fact}}^{\text{expected}})$$

6. Optionally, fine-tune $k_1, k_2, \Delta_1$ empirically to balance F1 and TPS.

| ID | $\tau_{\text{token}}$ | $\tau_{\text{seg}}^{\text{low}}$ | $\tau_{\text{seg}}^{\text{high}}$ | $\tau_{\text{global}}$ | F1 | TPS (s) |
|----|------|------|------|------|-------|--------|
| 1 | 0.30 | 0.52 | 0.70 | 0.68 | 42.17 | 125.34 |
| 2 | 0.30 | 0.54 | 0.72 | 0.70 | 43.89 | 132.58 |
| 3 | 0.30 | 0.55 | 0.74 | 0.71 | 44.36 | 138.45 |
| 4 | 0.40 | 0.55 | 0.75 | 0.70 | 51.03 | 113.29 |
| 5 | 0.40 | 0.57 | 0.77 | 0.72 | 49.03 | 110.87 |
| 6 | 0.40 | 0.58 | 0.79 | 0.73 | 50.21 | 118.46 |
| 7 | 0.50 | 0.60 | 0.80 | 0.74 | 52.45 | 140.12 |
| 8 | 0.50 | 0.62 | 0.82 | 0.75 | 53.18 | 148.77 |
| 9 | 0.50 | 0.63 | 0.84 | 0.76 | 54.09 | 156.92 |

Table 10: Illustrative threshold propagation from token to segment and global levels, with F1 and TPS. Row 4 highlights the high-performance, cost-efficient configuration.

### H.3 SELECTION OF TEMPERATURES

We provide a brief justification for the choice of a softmax temperature of 0.3 and a sampling temperature of 0.4 in Token-Guard.

Token-Guard depends on stable token-level hidden-state trajectories to perform reliable self-verification. High temperatures introduce excessive stochasticity during decoding, destabilizing these trajectories and weakening the consistency signals that Token-Guard relies upon. Lower temperatures, in contrast, smooth local transition dynamics and preserve the fine-grained patterns needed for hallucination detection.

Our choice of 0.3–0.4 is also aligned with the findings of Li et al. (2025), who show that low-temperature decoding (around 0.4) yields more stable distributional alignment in self-consistency reasoning, whereas higher temperatures distort the answer distribution and degrade consistency.

To further validate this choice, we conducted a small-scale sensitivity analysis across five temperature pairs. The 0.3–0.4 configuration achieved the best balance between hidden-state stability and output reliability; higher temperatures consistently reduced Token-Guard's ability to detect early hallucination drift.

| Softmax / Sampling | HaluEval | RAGTruth |
|--------------------|----------|----------|
| 0.15 / 0.25 | 75.63 | 41.94 |
| 0.30 / 0.40 | 79.56 | 43.60 |
| 0.50 / 0.60 | 73.34 | 43.31 |
| 0.80 / 0.80 | 67.05 | 42.30 |

Table 11: F1 performance under different temperature settings on HaluEval and RAGTruth.

Based on both theoretical considerations and empirical evidence, we therefore adopt 0.3 (softmax) / 0.4 (sampling) as the default temperature setting.

## I ERROR CASE STUDIES

This appendix presents five representative failure cases from context-dependent QA datasets, illustrating why some generations remain incorrect even after token-level self-checking and re-generation. Each case includes the dataset source, question, context, Token-Guard output, ground-truth answer, and a brief analysis of the failure reason.

**Case 1 – CovidQA**
**Question:** [In addition to oral swabs, which tests detected the presence of 2019-nCOV virus?]
**Context:** [Detectable 2019-nCoV viral RNA in blood is a strong indicator for the further clinical

severity. The study retrospectively analyzed virus presence in pharyngeal (oral) swabs, blood, and anal swabs in patients confirmed with 2019-nCoV. In the blood cohort, 6 of 57 patients had detectable RNA, all progressed to severe symptoms. In the anal swab cohort, 11 of 28 patients were positive, with 8 developing severe disease. Some patients had sequential samples across multiple days showing fluctuating viral loads. Patient 1 had detectable viral RNA in pharyngeal swabs on days 5, 7, 8, and 11; blood was weakly positive on day 6; anal swab was high on day 13. Patient 2 had higher viral RNA in anal swabs than blood on the same day. Patients 3-6 also showed varying RNA detection in blood and anal swabs corresponding with progression to severe illness. The study highlighted the correlation of extra-pulmonary viral RNA presence with disease severity, suggesting monitoring blood and anal samples could aid early prediction. Furthermore, high RNA in anal swabs indicates potential replication in the digestive tract. Limitations included missing stool samples and inability to measure bronchoalveolar lavage fluid due to procedural risks. Overall, results confirm virus RNA in extra-pulmonary sites and support early monitoring beyond oral swabs.]
**Token-Guard Output:** [Blood and anal swab samples]
**Ground Truth:** [The 2109-nCoV RNA was readily detected in the blood (6 of 57 patients) and the anal swabs (11 of 28 patients).]
**Analysis:** Token-Guard correctly identifies the sample types but omits the numeric details and sequential measurements. The failure stems from the challenge of summarizing long-form, densely detailed text into a complete answer, illustrating the common failure mode of generating accurate but abbreviated outputs for long, complex contexts.

**Case 2 – DROP_nfl**
**Question:** [How many points did each team have at halftime?]
**Context:** [Hoping to rebound from their tough overtime road loss to the Raiders, the Jets went home for a Week 8 duel with the Kansas City Chiefs. In the first quarter, New York took flight as QB Brett Favre completed an 18-yard TD pass to RB Leon Washington. In the second quarter, the Chiefs tied the game as QB Tyler Thigpen completed a 19-yard TD pass to TE Tony Gonzalez. The Jets would ground_truth with Washington getting a 60-yard TD run. Kansas City closed out the half as Thigpen completed an 11-yard TD pass to WR Mark Bradley.]
**Token-Guard Output:** [There isn't explicit information about the points scored by each team at halftime. However, two touchdowns have been scored by each team in the first two quarters: Washington's 18-yard TD pass and the 60-yard TD run, and Gonzalez's 19-yard TD pass and Bradley's 11-yard TD pass. Considering that each touchdown is worth 6 points, at halftime the scores are Jets 14 and Chiefs 12.]
**Ground Truth:** [14]
**Analysis:** The failure arises from the inherent difficulty of producing precise numerical answers when the context requires multi-step reasoning. Token-Guard correctly identifies the touchdowns but does not apply exact scoring rules to match the reference answer, illustrating a limitation in handling arithmetic reasoning within long contexts.

**Case 3 – PubMedQA**
**Question:** [Is ultrasound equal to X-ray in pediatric fracture diagnosis?]
**Context:** [Ultrasound is currently not established for the diagnosis of fractures. The aim of this study was to compare ultrasound and X-ray beyond their use solely for the identification of fractures, i.e., for the detection of fracture type and dislocation for pediatric fracture diagnosis. Limb bones of dead young pigs served as a model for pediatric bones. The fractured bones were examined with ultrasound, X-ray, and CT, which served as the gold standard. 162 of 248 bones were fractured. 130 fractures were identified using ultrasound, and 148 using X-ray. There were some advantages of X-ray over ultrasound in the detection of fracture type (80 correct results using X-ray, 66 correct results using ultrasound). Ultrasound, however, was superior to X-ray for dislocation identification (41 correct results using X-ray, 51 correct results using ultrasound). Both findings were not statistically significant after adjustment for multiple testing.]
**Token-Guard Output:** [No. Ultrasound was not equal to X-ray in detecting fracture type, but was superior to X-ray in identifying dislocations in pediatric fracture diagnosis.]
**Ground Truth:** [Yes]
**Analysis:** This failure primarily arises from incomplete or ambiguous evidence in the context. Although the numerical differences exist between ultrasound and X-ray, the statistical non-significance indicates comparable effectiveness overall. Token-Guard overemphasizes minor differences in frac-

ture type detection, illustrating how subtle interpretations in biomedical contexts can lead to discrepancies even after re-generation.

**Case 4 – HaluEval**
**Question:** [Which genus of moth in the world's seventh-largest country contains only one species?]
**Context:** [Indogrammodes is a genus of moths of the Crambidae family. It contains only one species, Indogrammodes pectinicornalis, which is found in India. India, officially the Republic of India, is a country in South Asia. It is the seventh-largest country by area, the second-most populous country (with over 1.2 billion people), and the most populous democracy in the world.]
**Token-Guard Output:** [Indogrammodes]
**Ground Truth:** [Crambidae]
**Analysis:** This failure is due to ambiguous context. Both the genus *Indogrammodes* and its family *Crambidae* are mentioned, allowing multiple plausible answers. Token-Guard outputs the genus, which is factually correct but differs from the reference answer, illustrating errors from ambiguous grounding.

**Case 5 – DROP_history**
**Question:** [What religion was least prominent in 2008?]
**Context:** [Greater Manchester is home to a diverse population and is a multicultural agglomeration with an ethnic minority population comprising 8.5% of the total population in 2001. In 2008, there were over 66 refugee nationalities in the county. At the 2001 UK census, 74.2% of Greater Manchesters residents were Christian, 5.0% Muslim, 0.9% Jewish, 0.7% Hindu, 0.2% Buddhist, and 0.1% Sikh. 11.4% had no religion, 0.2% had an alternative religion and 7.4% did not state their religion. This is similar to the rest of the country, although the proportions of Muslims and Jews are nearly twice the national average. It contains the Heaton Park Hebrew Congregation, a large Ashkenazi Orthodox synagogue in North Manchester. Greater Manchester is covered by the Roman Catholic Dioceses of Roman Catholic Diocese of Salford and Roman Catholic Diocese of Shrewsbury, and the Roman Catholic Archdiocese of Liverpool. Most of Greater Manchester is part of the Anglican Diocese of Manchester, apart from Metropolitan Borough of Wigan which lies within the Anglican Diocese of Liverpool.]
**Token-Guard Output:** [Buddhist]
**Ground Truth:** [Sikh]
**Analysis:** The model misidentified the least prominent religion due to the subtle difference between 0.2% Buddhist and 0.1% Sikh in the context. This illustrates a failure arising from fine-grained numerical reasoning within the context, where token-level self-checking alone cannot fully resolve minute quantitative distinctions.

### I.1 Summary of Failure Patterns

Across these five representative cases, common failures arise from incomplete or ambiguous grounding in the context, where multiple plausible answers exist or key information is subtle. Some errors also stem from limitations in the model's knowledge, especially for rare or specialized topics.

Additional failures occur due to trade-offs between hallucination suppression and expressive completeness, leading to omitted details or numerical information, and from multi-step reasoning or long-form aggregation that exceeds token-level verification. These examples illustrate both the effectiveness of Token-Guard in reducing hallucinations and the practical limits of context-dependent QA under current model and dataset constraints.

## J Limitations and Future Work

While Token-Guard demonstrates strong performance across diverse datasets and backbone models, several limitations remain.

**First**, the multi-stage decoding with token-level and segment-level scoring introduces non-negligible computational overhead, particularly on longer passages, which may limit scalability on smaller models or latency-sensitive applications.

**Second**, Token-Guard primarily mitigates hallucinations during decoding but does not supplement missing knowledge; on tasks requiring extensive domain-specific knowledge, factual improvements are constrained by the backbone LLM. In particular, when the backbone LLM lacks key knowledge, hallucinations stem from missing information rather than decoding errors.

**Third**, the clustering and global iteration mechanisms assume semantically coherent segments; highly fragmented or noisy input can reduce effectiveness of segment-level scoring and chain selection. Segment scoring remains limited on messy or fragmented inputs, clarifying the boundary of Token-Guard's mitigation capabilities.

**Finally**, Token-Guard currently focuses on textual knowledge and structured reasoning chains; extending it to multi-modal inputs remains a promising direction.

