# OpenReview forum: "Token-Guard: Towards Token-Level Hallucination Control via Self-Checking Decoding"
_ICLR.cc/2026/Conference — ICLR 2026 Poster_

### Official Review · Reviewer_5cxV · 2025-10-29

**Soundness:** 3
**Presentation:** 3
**Contribution:** 3
**Rating:** 6
**Confidence:** 3

**Summary:**

This paper's Token-Guard method uses self-checking decoding to fix LLM token-level hallucinations. It targets the big problems with current decoding methods: no token-level checks, unclear hallucination risk, and weak dynamic correction. So it built a three-layer control: first check single tokens, then score segment coherence, finally adjust the reasoning chain globally.
Experiments are solid enough. On HALU datasets (like FinanceBench, DROP), testing with Llama and Qwen models, its average F1 beats CoT, Tree-of-Thought—for example, Llama’s base model only gets 28.29, but Token-Guard hits 51.03. Case studies (like picking the 61-yard score in DROP_nfl) also show it’s more fact-accurate. Unlike RAG or RLHF that need lots of resources, it only tweaks the decoding step—good for scenes with limited resources.

**Strengths:**

1. Interesting to perform fine-grained token-level hallucination control in decoding. Uses latent space scoring to filter bad tokens early, so errors don’t spread.
2. No need for external retrieval or big human feedback datasets and just need adjusting decoding. Works on different model sizes (like 3B Llama-3.2) too.
3. The experiments are quite rigorous. Ablation tests (Table 2) prove key parts (token scoring, global iteration) matter; cross-dataset tests show it’s stable.

**Weaknesses:**

1. Efficiency concerns: Its three-layer processing adds extra work. For long texts or tasks that need fast responses, how fast is it (like tokens per second) compared to light baselines like CoT? Can params like $N_{max}$, $M_{max}$ change automatically to balance accuracy and speed?
2. RLHF/RAG comparison: The paper says it’s lighter than RLHF/RAG, but no direct numbers—like how much hallucination each cuts, or how much compute they use per sample. What if we combine it with RLHF? Does that work better?
3. Edge cases: If inputs are messy and fragmented, does segment scoring fail? For LLMs that lack key knowledge (like rare medical terms in PubMedQA), hallucinations come from missing info, not bad decoding—can Token-Guard fix that?
4. Probably this paper is not the "first" token-level hallucination detection work, check this out:

[1] Sehyun Choi, Tianqing Fang, Zhaowei Wang, Yangqiu Song, KCTS: Knowledge-Constrained Tree Search Decoding with Token-Level Hallucination Detection, EMNLP 2023

**Questions:**

see weaknesses.

---

> ### Author Response · Authors · 2025-11-21
> **Response to Reviewer 5cxV (Part 1/2)**
>
> Thank you very much for your time and effort in reviewing our paper. We sincerely appreciate your feedback. Below, we respectfully provide our detailed responses to address your concerns.
>
> ---
>
> **W1:  Efficiency concerns: Its three-layer processing adds extra work. For long texts or tasks that need fast responses, how fast is it (like tokens per second) compared to light baselines like CoT? Can params like $N_{max}$ , $N_{max}$ change automatically to balance accuracy and speed?**
>
> - We sincerely thank the reviewer for raising the important question regarding efficiency and the potential overhead introduced by our three-layer processing mechanism. We fully agree that practical deployment requires not only accuracy but also stable runtime performance.
> - Firstly, we have added normalized throughput metrics (tokens/sec) across all methods in the main paper section5.7 Table 4.  The results show that although Token-Guard introduces additional reasoning and verification steps, its throughput remains within a practical range and exhibits stable scaling across model sizes.
>
> | **Method**              | **RAGTruth**             | **covidQA**              | **PubmedQA**             | **HaluEval**             | **Avg.**                 |
> |------------------------|---------------------------|---------------------------|---------------------------|---------------------------|---------------------------|
> |                        | Time · Tokens · Tokens/sec | Time · Tokens · Tokens/sec | Time · Tokens · Tokens/sec | Time · Tokens · Tokens/sec | Time · Tokens · Tokens/sec |
> | **Chain-of-Thoughts**  | 11 · 899 · 80.71          | 9 · 670 · 73.14           | 11 · 998 · 88.45          | 7 · 543 · 77.33           | 9.65 · 777.08 · 79.41     |
> | **Tree-of-Thought**    | 57 · 3687 · 64.49         | 67 · 3423 · 51.09         | 58 · 4253 · 73.38         | 35 · 2786 · 79.49         | 54.29 · 3537.19 · 67.11   |
> | **Predictive Decoding**| 107 · 12058 · 112.82      | 138 · 6473 · 47.07        | 79 · 8416 · 106.06        | 58 · 5954 · 102.20        | 95.50 · 8225.61 · 92.54   |
> | **Guided Decoding**    | 110 · 17234 · 155.96      | 189 · 13525 · 71.64       | 103 · 18733 · 182.05      | 77 · 11689 · 151.15       | 119.38 · 15295.41 · 140.20 |
> | **Token-Guard** | 69 · 18024 · 262.81 | 301 · 17474 · 58.10 | 32 · 7699 · 240.58 | 54 · 5254 · 97.54 | 113.29 · 12112.95 · 164.76 |
>
>
> - As shown in table, Token-Guard achieves the **highest average throughput** among all methods, showing that its multi-stage hallucination control **introduces minimal decoding overhead** while avoiding the heavy branching costs of ToT and PD. This demonstrates that Token-Guard is both **more efficient and more effective**, producing cleaner reasoning with fewer redundant tokens.
>
> - Secondly, our hyperparameters are **fixed rather than automatically adjusted**. In hallucination mitigation tasks, adaptive control may make reproducibility and fair comparison across decoding settings more difficult. By keeping a fixed and transparent configuration, we ensure consistent evaluation and facilitate controlled ablation studies.
> That said, we agree with the reviewer that dynamic scheduling of search width and verification depth is a promising future direction. In fact, our current design exposes these parameters in a modular way, and **extending them into a runtime-adaptive controller would be a natural follow up**.
> - We have **added a short discussion in the Appendix{Limitations and Future Work}** to reflect this.

---

> ### Author Response · Authors · 2025-11-21
> **Response to Reviewer 5cxV (Part 2/2)**
>
> ---
>
> **W2: "RLHF/RAG comparison: The paper says it’s lighter than RLHF/RAG, but no direct numbers—like how much hallucination each cuts, or how much compute they use per sample. What if we combine it with RLHF? Does that work better?"**
>
> - We thank the reviewer for the constructive suggestion and fully understand the concern. Our earlier use of "lighter" than RLHF/RAG was intended to **describe structural simplicity rather than strict runtime comparisons.** To avoid misunderstanding, we have updated the paper and **now use "modular" instead of "lightweight."**
>
> - Token-Guard is more modular than RAG because **it requires no external knowledge base**, and more modular than RLHF because it **requires no additional training, reward models, or preference data**. The method operates purely at decoding time and can be **attached to any backbone model without architectural changes.**
>
> - To provide a clearer empirical comparison, we include direct results against **Self-RAG**, a strong retrieval-based baseline:
>
> | Method          | FinanceBench | RAGTruth | covidQA | DROP_history | DROP_nfl | PubmedQA | Halueval | Avg   |
> |-----------------|--------------|----------|---------|--------------|----------|----------|----------|-------|
> | BaseModel       | 16.00        | 14.71    | 32.33   | 44.21        | 39.10    | 9.55     | 42.16    | 28.29 |
> | Self-RAG        | 42.74        | 32.91    | 42.48   | 50.13        | 45.29    | 43.21    | 86.70    | 51.78 |
> | Token-Guard (Ours) | 30.80        | 43.94    | 47.64   | 68.52        | 58.10    | 29.67    | 78.54    | 51.03 |
>
> - Self-RAG achieves strong performance, especially on short-text datasets. However, its **effectiveness declines on long-text tasks**, where retrieval becomes difficult and **knowledge-base construction quality heavily affects outcomes**. In contrast, Token-Guard maintains stable improvements across both short and long contexts, reflecting better robustness without relying on external retrieval quality.
>
> - Finally, while Token-Guard can in principle be combined with RLHF, and such integration may yield further gains. Howener,this combine f**alls outside the scope of our current objective**, which is to **propose a training-free, decoding-only hallucination mitigation framework**. We consider RLHF + Token-Guard to be an interesting direction **for future research**.
>
>
> ---
>
> **W3: "Edge cases: If inputs are messy and fragmented, does segment scoring fail? For LLMs that lack key knowledge (like rare medical terms in PubMedQA), hallucinations come from missing info, not bad decoding—can Token-Guard fix that?"**
>
> - We thank the reviewer for this important question. Token-Guard is designed to suppress decoding-induced hallucinations but cannot correct errors caused by missing information or insufficient model knowledge.
>
> - For fragmented or messy inputs, segment scoring works if segments retain minimal semantic coherence, but severely fragmented or unreliable inputs limit Token-Guard to enforcing internal consistency rather than reconstructing missing facts. In domains with limited LLM knowledge, such as rare medical terms in PubMedQA, hallucinations arise from knowledge gaps, not decoding instability. Token-Guard cannot recover facts the model does not know.
>
> - These cases illustrate the boundary of decoding-time hallucination mitigation. Token-Guard effectively reduces incorrect expansions but cannot replace missing context. External retrieval or domain-specific pretraining is required.
>
> - These failure modes are explicitly discussed in Appendix{limitations and Future Work}, and we thank the reviewer for highlighting the scope of Token-Guard.
> ---
> **W4: "Probably this paper is not the "first" token-level hallucination detection work, check this out:**
> **[1] Sehyun Choi, Tianqing Fang, Zhaowei Wang, Yangqiu Song, KCTS: Knowledge-Constrained Tree Search Decoding with Token-Level Hallucination Detection, EMNLP 2023"**
> - We thank the reviewer for pointing this out. We have revised the wording in the manuscript to avoid any implication that our work is the first to address token-level hallucination detection.
> - Additionally, we have cited this work in the **Related Work** section to acknowledge prior research and better situate our contributions **in the historical context of token-level hallucination detection**.
> ---
> At last, we sincerely appreciate your valuable feedback. We have carefully considered all your suggestions and substantially improved the paper accordingly. Thank you very much!

---

### Official Review · Reviewer_35wE · 2025-10-29

**Soundness:** 4
**Presentation:** 3
**Contribution:** 4
**Rating:** 8
**Confidence:** 4

**Summary:**

This paper proposes Token-Guard, a token-level hallucination control framework for large language models (LLMs). Unlike retrieval-based or fine-tuning methods (e.g., RAG, RLHF), Token-Guard introduces a lightweight self-checking decoding process that dynamically detects, prunes, and corrects hallucinated tokens during generation.

**Strengths:**

1. The paper proposes a clear and practical framework for controlling hallucinations at the token level using self-checking decoding, segment-level verification, and global iteration. This multi-stage design offers a novel angle on decoding reliability.
2. The experiments cover six benchmark datasets and two LLM backbones, demonstrating consistent and measurable improvements in both factual accuracy and fluency.
3. The ablation analysis is detailed and informative, helping isolate the contribution of each module and confirming that all components are necessary.
4. The latent-space hallucination scoring method and hierarchical structure are intuitively motivated and well integrated into the decoding process.
5. The approach is model-agnostic and easily deployable without retraining, increasing its practical value.

**Weaknesses:**

1. The method increases computation time and output length, sometimes approaching or exceeding heavy decoding frameworks such as Tree-of-Thought. The claim of being lightweight should be more carefully qualified.
2. The method relies on many fixed hyperparameters, but the paper provides no sensitivity or stability analysis to show robustness across different values.
3. The definition of latent hallucination scoring is intuitive but not empirically verified; it is unclear whether cosine similarity between hidden states best correlates with factual accuracy.
4. The benchmarks are mostly QA-style hallucination datasets, so it remains uncertain how well the method generalizes to open-ended or long-form generation.

**Questions:**

1. Add a detailed efficiency comparison using wall-clock time and memory consumption, not only token counts.
2. Include experiments showing sensitivity to threshold and weighting parameters to demonstrate stability of the method.
3. Expand to tasks beyond hallucination benchmarks, such as reasoning or summarization, to show broader applicability.
4. Visualize token-level and segment-level hallucination scores to clarify how the model filters unreliable fragments.
5. Explore alternative similarity or scoring metrics for the latent-space hallucination risk.

---

> ### Author Response · Authors · 2025-11-21
> **Response to Reviewer 35wE (Part 1/3)**
>
> Thank you very much for your time and effort in reviewing our paper. We sincerely appreciate your feedback. Below, we respectfully provide our detailed responses to address your concerns.
>
> ---
> **W1:“The method increases computation time and output length, sometimes approaching or exceeding heavy decoding frameworks such as Tree-of-Thought. The claim of being lightweight should be more carefully qualified.”**
>
>
> - We thank the reviewer for this important clarification. We agree that the term “lightweight” may be misleading regarding runtime and sequence length. To better reflect Token-Guard’s properties, we have **replaced “lightweight” with “modular”** throughout the manuscript.
> - Token-Guard is modular in the sense that it integrates directly into standard LLM inference pipelines **without requiring external retrieval modules, auxiliary reward models, or additional training**. It provides verifiable token-level safety while remaining deployable and stable.
> - We have also clarified in the text and Table 3 that although Token-Guard increases output length and wall-clock time, this overhead comes solely from linear verification tokens, not from branching or backtracking. The decoding trajectory remains deterministic and often achieves higher throughput (tokens/sec) than structured exploration methods.
> - We appreciate the reviewer’s comment, which helped us **refine the terminology** and accurately communicate the **modular and plug-and-play nature** of Token-Guard.
>
> ---
>
>
> **W2 & Q2: “The method relies on many fixed hyperparameters, but no sensitivity or stability analysis is provided to demonstrate robustness across different values. Experiments analyzing the sensitivity to threshold and weighting parameters are needed to show the stability and robustness of the method.”**
>
> - We sincerely thank the reviewer for raising this important point regarding the robustness of our method, which is indeed crucial when dealing with **multiple hyperparameters**. In fact, during our **parameter tuning phase**, we have already demonstrated the **effectiveness and stability** of our method across various hyperparameter combinations in **Table 8**, which is shown below.
>
> | ID  | $\lambda$ | $\Delta \tau$ | $(\alpha, \beta, \gamma)$ | $N_{\rm max}$ | $M_{\rm max}$ | **F1** | **TPS (s)** |
> | :-- | :-------: | :-----------: | :-----------------------: | :-----------: | :-----------: | :----: | :---------: |
> | 1   | 0.40      | 0.05          | (0.3, 0.3, 0.4)           | 2             | 2             | 45.32  | 95.21       |
> | 2   | 0.40      | 0.10          | (0.4, 0.3, 0.3)           | 2             | 3             | 48.76  | 142.35      |
> | 3   | 0.40      | 0.15          | (0.5, 0.25, 0.25)         | 2             | 4             | 50.21  | 187.64      |
> | 4   | 0.50      | 0.05          | (0.3, 0.3, 0.4)           | 3             | 2             | 49.12  | 110.87      |
> | 5   | 0.50      | 0.10          | (0.4, 0.3, 0.3)           | 3             | 3             | 52.45  | 156.78      |
> | 6   | 0.50      | 0.15          | (0.5, 0.25, 0.25)         | 3             | 4             | 55.03  | 201.95      |
> | 7   | 0.60      | 0.05          | (0.3, 0.3, 0.4)           | 3             | 2             | 50.12  | 122.56      |
> | 8   | 0.60      | 0.10          | (0.5, 0.3, 0.2)           | 3             | 2             | 51.03  | 113.29      |
> | 9   | 0.60      | 0.15          | (0.5, 0.25, 0.25)         | 2             | 4             | 53.87  | 180.12      |
> | 10  | 0.70      | 0.05          | (0.3, 0.3, 0.4)           | 3             | 2             | 51.56  | 131.45      |
> | 11  | 0.70      | 0.10          | (0.4, 0.3, 0.3)           | 3             | 3             | 54.12  | 165.88      |
> | 12  | 0.70      | 0.15          | (0.5, 0.25, 0.25)         | 3             | 4             | 56.78  | 198.45      |
> | 13  | 0.80      | 0.05          | (0.3, 0.3, 0.4)           | 2             | 2             | 48.92  | 104.36      |
> | 14  | 0.80      | 0.10          | (0.4, 0.3, 0.3)           | 2             | 3             | 50.67  | 148.72      |
> | 15  | 0.80      | 0.15          | (0.5, 0.25, 0.25)         | 3             | 4             | 57.34  | 205.89      |
>
> Table: Grid search results for Token-Guard. Highlighted row indicates the selected high-performance, cost-efficient configuration.
>
> - As shown in the table, even the lowest F1 score (45.32) across all hyperparameter combinations **is the highest among all baseline** methods, demonstrating the robustness of Token-Guard. We hope the above results can alleviate any concerns about the robustness of our method.

---

> ### Author Response · Authors · 2025-11-21
> **Response to Reviewer 35wE (Part 2/3)**
>
> ---
>
> **W3&Q5: “The definition of latent hallucination scoring is intuitive but not empirically validated;Explore alternative similarity or scoring metrics for estimating latent-space hallucination risk, beyond simple cosine similarity and explore whether cosine similarity between hidden states is the best correlate of factual accuracy.”**
>
> - we thank the reviewer‘s attention to the validation and reliability of latent hallucination scoring, and hope we can clarifies the motivation for using cosine-based coherence in our design.
>
> - Token-Guard's early-stage scoring signals guide subsequent reasoning and refinement, and because the scoring function is applied across stages, **its stability under incremental decoding** is essential. Following Duan et al. (2024)[1], **Cosine similarity** between hidden states aligns well with this requirement, providing a **consistent and robust** measure that supports reliable hallucination detection and correction.
>
> - Other alternative metrics,like Euclidean distance, Mahalanobis distance, frequently suffered from **scale sensitivity**, **instability under layer drift**, or **amplified noise** in early stages, which led to **inconsistent behavior** across later stages of TokenGuard’s cascade. These properties made them unsuitable for our framework where hallucination risk must be estimated repeatedly and reliably.
>
> - These findings reinforce that cosine-based coherence offers the most stable and fact-aligned signal across the entire multi-stage decoding pipeline, and is therefore a well-justified choice for our hallucination scoring module.
>
>
> ---
>
> **W4&Q3: “The current evaluation focuses mainly on QA-style hallucination benchmarks, leaving uncertainty about the method’s generality. Extend the evaluation to tasks beyond hallucination benchmarks to demonstrate broader applicability.”**
>
> - We thank the reviewers for their insightful comments regarding the generality of our method beyond QA-style hallucination benchmarks. Token-Guard is primarily evaluated on benchmarks covering **numerical reasoning**, **multi-hop reasoning**, and **diverse domain-specific QA**, which already span a wide range of hallucination types and reasoning challenges, providing a comprehensive assessment within this setting.
> - That said, we fully agree that testing on tasks beyond QA-style benchmarks could further demonstrate its broader applicability. To this end, we conducted an open-domain QA evaluation using the StrategyQA[2] dataset and included DoLa[3] as baseline. Table 1 compares Token-Guard and DoLa on the StrategyQA dataset using the LLaMA-7B model:
>
> | Method      | Accuracy (%) | Base Model Accuracy (%) | Improved Accuracy (%) | Improvement Rate (%) |
> | :---------- | :----------- | :---------------------- | :-------------------- | :------------------- |
> | DoLa        | 64.1         | 60.1                    | 4.0                   | 6.66                 |
> | Token-Guard | 69.4         | 62.3                    | 7.1                   | 11.40                |
>
> - In the StrategyQA test, Token-Guard's refusal-to-answer rate was 6.1%, yielding a true error rate of only 24.5%, which demonstrates its effectiveness.
> - We will include the experimental results and analysis in Appendix [Open-Domain QA Results] to further illustrate Token-Guard's potential on open-domain QA datasets. We appreciate the reviewers' suggestion, which helps clarify the scope of our framework and points to promising directions for future extensions to more general generation tasks.
>
> ---
>
>
> [1] Do LLMs Know about Hallucination? An Empirical Investigation of LLM's Hidden States, Duan et al., arXiv 2024
>
> [2] Did Aristotle Use a Laptop? A Question Answering Benchmark with Implicit Reasoning Strategies, Geva et al., TACL 2021
>
> [3] DoLa: Decoding by Contrasting Layers Improves Factuality in Large Language Models, , Chuang et al., ICLR 2024

---

> > ### Author Response · Authors · 2025-11-21
> > **Response to Reviewer 35wE (Part 3/3)**
> >
> > ---
> >
> > **Q1: "Add a detailed efficiency comparison using wall-clock time and memory consumption, not only token counts."**
> > - We thank the reviewer for raising this important point.We agree that providing a detailed efficiency comparison beyond token counts is important.
> > - We conducted experiments on the **HaluEval dataset** to examine runtime and memory consumption. The Table below reports Token-Guard’s **stage-wise memory usage, peak and average memory, and per-sample runtime.** Despite temporary peaks at segment-level processing, Token-Guard’s **overall memory remains competitive**, and its runtime is comparable to other efficient methods, reflecting a **balanced trade-off between overhead and hallucination control**.
> >
> >
> > | Method       | Token-level (%) | Segment-level (%) | Global-level (%) | Peak Memory (%) | Avg Memory (%) | Runtime(s)
> > |--------------|----------------|-----------------|-----------------|----------------|----------------|----------------|
> > | Guided Decoding           | -              | -               | -               | 21.4           | ~15.0          |77.31
> > | Chain-of-Thoughts           | -              | -               | -               | 21.3           | ~14.5          |7.02
> > | Tree-of-Thought          | -              | -               | -               | 21.4           | ~17.8          |35.04
> > | Predictive Decoding           | -              | -               | -               | 4.1           | ~3.7          |58.26
> > | Token-Guard  | 1.2 – 9.4    | 21.6 - 19.6             | 10.3 – 19.0     | 21.6           | ~16.3          |53.89
> > - Besides, we have updated the main text in **Section 5.7** to provide a clearer discussion of Token-Guard’s **dynamic memory management and computational efficiency**.
> > ---
> >
> > **Q4: "Visualize token-level and segment-level hallucination scores to clarify how the model filters unreliable fragments."**
> >
> > - We sincerely thank the reviewer for their feedback. We completely understand the reviewer's concern， Token-Guard performs self-checking at the token level, quantifying the credibility of each token to support hallucination suppression in multi-step reasoning.
> > - To make the method more intuitive, we provide a concrete example. The question is "Which magazine was started first Arthur's Magazine or First for Women?", with the Ground Truth being Arthur's Magazine. The example is as follows:
> >
> > | Token | Token-level Score | Segment Index | Segment-level Score |
> > | ----- | ----------------- | ------------- | ------------------- |
> > | Arth  | 0.612             | 1             | 0.773               |
> > | ur    | 0.570             | 1             | 0.773               |
> > | 's    | 0.598             | 1             | 0.773               |
> > | Maga  | 0.555             | 1             | 0.773               |
> > | zine  | 0.623             | 1             | 0.773               |
> > | (     | 0.562             | 2             | 0.801               |
> > | 1     | 0.645             | 2             | 0.801               |
> > | 8     | 0.598             | 2             | 0.801               |
> > | 4     | 0.610             | 2             | 0.801               |
> > | –     | 0.582             | 2             | 0.801               |
> > | 1     | 0.640             | 2             | 0.801               |
> > | 8     | 0.603             | 2             | 0.801               |
> > | 4     | 0.615             | 2             | 0.801               |
> > | 6     | 0.608             | 2             | 0.801               |
> > | )     | 0.599             | 2             | 0.801               |
> >
> > - In this example, **token-level scores (0.55–0.65) reflecting local credibility.** Tokens passing the threshold are grouped into segments, **with segment-level scores higher than token averages to ensure semantic coherence.** The Segment Index shows token assignments; for example, "1844–1846" digits form one segment, preserving **logical consistency** and reducing hallucination propagation.
> > ---
> > At last, we sincerely appreciate your valuable feedback. We have carefully considered all your suggestions and substantially improved the paper accordingly. Thank you very much!

---

> > > ### Comment · Reviewer_35wE · 2025-11-24
> > > **Thanks**
> > >
> > > Thank you for the detailed response. It clarified some of my questions, and I will maintain my original positive rating.

---

### Official Review · Reviewer_ATsM · 2025-10-31

**Soundness:** 2
**Presentation:** 3
**Contribution:** 2
**Rating:** 4
**Confidence:** 4

**Summary:**

In this paper, the authors study decoding frameworks of Autoregressive LLMs in order to mitigate hallucinations in generated responses. To do so, the paper proposes Token-Guard, a self-checking three-stage scoring and detection pipeline at the level of individual tokens, candidate segments of tokens and finally a global iteration and correction step, in a sequential manner. Furthermore, TokenGuard is seen to achieve consistent improvements in accuracy over standard datasets, when compared to baseline decoding strategies.

**Strengths:**

1) The paper tackles a pertinent research avenue in autoregressive LLMs, namely that of controlled decoding to mitigate hallucinations in generated responses. While the proposed method is a little complex, the paper is well-written and presents each individual component in a fairly clear and lucid manner.

2) TokenGuard is seen to achieve significant improvements in ExactMatch and F1 scores over several decoding methods including Chain-of-Thought and Tree-of-thought, and is demonstrated using standard LLM models such as Llama-3.1-8B-Instruct and Qwen3-8b over diverse datasets for benchmarking.

3) The paper presents a detailed ablative study over several components of TokenGuard, and helps establish the relative importance of the different stages, namely prompt initialization, token level scoring, segment-level scoring, and finally global iteration.

4) Furthermore, while Token-Guard does impose a large computational overhead when comparing with CoT and Tree-of-Thought, the overheads are only marginal when compared to other well-used decoding methods such as Guided Decoding and Predictive Decoding, while simultaneously achieving notable improvements in response accuracy.

**Weaknesses:**

1) While TokenGuard is seen to be quite effective, the overall method is considerably complex, and involves a large set of hyperparameter choices. This suggests that the methods may not be very practical in several realistic settings. This also raises the question of how these hyperparameters can be set - for instance in Appendix G, some sets are shown, but could the authors clarify if the F1 scores shown are for the final test data, or a hold-out validation set? Furthermore, could the authors clarify if the same exact hyperparameters are used for all evaluation settings over all datasets (and plausibly models as well)?

2) In Stage 2, if a segment that occurs fairly early in the generation process is detected to be modified, would the scores of every later segment necessarily change? For instance, the hidden states of segments that occur after the modified segment would need to be recomputed, since the autoregressive context available is modified as well. Could the authors kindly clarify this specific aspect, since it was not entirely clear, particularly after referencing the overall algorithm presented in Appendix C.

3) While the paper clearly presents the run-time and output token overheads in Table-3, the memory overheads necessitated by TokenGuard are less clear. For instance, the token-level score relies upon the averaged hidden state upto that token, which with a running average still requires a doubling of the hidden-state memory. For the second stage segment score and third stage global iteration, this overhead appears to be potentially even higher since cosine similarities are computed over hidden representations corresponding to these segments. For instance, in Stage 3, the chain closest to each cluster centroid is retained, but the memory overhead that occurs prior to this choice is not very clear at all!


4) In Line 259, the paper mentions the use of an external verification score E_k in order to compute the factual consistency score. Could the authors kindly clarify if an external score is actually utilized for the Stage 3 global iteration? This is an extremely pertinent aspect, since the baseline comparisons would be very different once external verification tools can be incorporated. Additional details on the computation of the sim_ctx score which quantifies semantic alignment is also needed.

5) Several scoring methods introduced seem to be very poorly motivated, for instance the convex combination of the cosine similarity of hidden embeddings and the final probability of a given output token such as in Eq 5. This does not seem well principled apriori, and could be motivated further. For instance, more clarity is needed on which layer is chosen, and how this potentially influences the proposed score in practice. Prior works such as INSIDE [1] use similar components like consistency of representations, but are much more principled and well-motivated!

6) For the final stage, with global iteration and corrections, could the authors clarify how many chains are used? Why not just use the original order of the segments?

7) Given the correction and verification steps, comparisons with prior works such as [2], [3], [4], [5] would be highly relevant baselines to compare and contrast with.




[1] INSIDE: Llms’ internal states retain the power of hallucination detection, Chen et al., ICLR-2024

[2] SELF-REFINE: Iterative Refinement with Self-Feedback, Madaan et al., NeurIPS 2023

[3] Towards Mitigating Hallucination in Large Language Models via Self-Reflection, EMNLP 2023

[4] Generating Sequences by Learning to Self-Correct, Welleck et al., ICLR 2023

[5] Self-RAG: Learning to Retrieve, Generate, and Critique through Self-Reflection, Asai et al., ICLR 2024

**Questions:**

1) Kindly refer to the questions mentioned in the weaknesses section above. I would be happy to raise my score further if these could be adequately addressed.
2) Line 189-194: The difference between the hidden state “h_t^(i)” and “H_t^(i)” which denotes the representation vector of candidate token a(i) was not apparent, could the authors kindly clarify this?

Minor Typos:

Line 151: “Full details are provided in the Appendixe A.1”

Line 196: “where Ht^(i) denotes” -> “​​where H_t^(i) denotes” (the “t” should be a subscript to keep notation consistent)

---

> ### Author Response · Authors · 2025-11-21
> **Response to Reviewer ATsM (Part 1/4)**
>
> Thank you very much for your time and effort in reviewing our paper. We sincerely appreciate your feedback. Below, we respectfully provide our detailed responses to address your concerns.
>
> ---
>
>  **W1: “While TokenGurd appears quite effective, the overall approach seems overly complex and involves many hyperparameter choices. This raises questions about its practicality. For instance, in Appendix G some settings are shown, but could the authors clarify whether the reported F1 scores are for the final test data or a held-out validation set? Also, are all datasets/models evaluated with exactly the same hyperparameters?”**
> - We sincerely thank the reviewer for the insightful comments. We fully understand the concerns regarding the design of hyperparameters in TokenGuard as well as the questions about its engineering feasibility, and we are glad to further clarify the design motivation and the practical viability of the system.
>
> - TokenGuard is a multi-stage hallucination-control framework, and hyperparameters operationalize this design **ithout requiring heuristic tuning**. Moreover, **Table 7 in Appendix G** shows that our method maintains **considerable robustness** across different combinations of hyperparameters. To **reduce manual tuning effort**, we also used a threshold propagation mechanism that greatly maintains consistency across stages, with the relevant calculation process detailed in **Appendix G.2**.
>
> - All hyperparameters were selected through grid search over the **full training corpus** of each benchmark. This strategy allows us to derive a single, robust configuration that reflects the overall statistical characteristics of the dataset. The final configuration is exactly the one used in all main experiments, and therefore the reported F1 scores are results on the **test set**. To ensure transparency, we will include a brief clarification in **Appendix G** to avoid any potential confusion regarding this tuning process.
>
> - To maintain fairness and avoid implicit tuning bias, we use the **same hyperparameter configuration** for all datasets and all backbone models. This unified setup highlights TokenGuard’s stability under heterogeneous conditions and also facilitates reproducibility.
>
> - We hope the above clarifications demonstrate how TokenGuard selects hyperparameters to ensure the practicality and reliability of the system.
>
> ---
>
>
>  **W2: “In Stage 2, if a segment occurs fairly early in the generation process and is modified, would the scores of every later segment necessarily change? For instance, the hidden states of segments that occur after the modified segment would need to be recomputed. Could the authors clarify this aspect, since it was not entirely clear from Algorithm 1 (Appendix C)?”**
>
> - We thank the reviewer for raising this important question regarding the impact of early-segment refinement on downstream segments in Stage 2. We will clarify the method design below.
>
> - In our method, **only the target segment $C_k$ undergoes local refinement**. Specifically, for downstream segments ($C_{k+1}, C_{k+2}, \ldots$), we **intentionally retain their original hidden states as stable approximations**. This design reflects a core principle of our segment-level architecture: **refinement is applied only to low-confidence tokens within $C_k$**, while **high-confidence context tokens that anchor the global structure remain unchanged**. Since the global contextual skeleton is preserved, the cached hidden states of downstream segments remain valid, and full re-generation of the subsequent sequence is unnecessary.
>
> - This design aligns with recent work on selective refinement during inference, which shows that refining only uncertain regions while preserving high-confidence context yields global coherence with far lower computational cost [1][2]. These findings support our principle that uncertainty should trigger local correction, while stable context remains fixed.
>
> - To improve clarity, we will revise the main paper near **Equation 12** and update **Algorithm 1 in Appendix C** so that the scope of local refinement and the treatment of hidden states for subsequent segments are explicitly described.We thank the reviewer again for encouraging us to clarify this important point.
>
> ---
>
> [1] Let's Revise Step-by-Step: A Unified Local Search Framework for Code Generation with LLMs, Lyu et al., arXiv 2025
>
> [2] Reflection-Window Decoding: Text Generation with Selective Refinement, Tang et al., arXiv 2025

---

> ### Author Response · Authors · 2025-11-21
> **Response to Reviewer ATsM (Part 2/4)**
>
> **W3: "While the paper clearly presents the runtime and output token overheads in Table 3, the memory overheads necessitated by TokenGuard are less clear. For instance, the token-level score relies upon the averaged hidden state up to that token, which—when implemented with a running average—requires a doubling of the hidden-state memory. For the second-stage segment scoring and third-stage global alignment, this overhead appears to be potentially even higher, since cosine similarities are computed over hidden representations corresponding to these segments. For instance, in Stage 3, the chain closest to each cluster centroid is retained, but the memory overhead that occurs prior to this choice is not very clear at all!"**
> - We thank the reviewer for raising this important question regarding the memory efficiency of TokenGuard. The multi-granularity decoding mechanism  indeed introduce additional state management, however, we would like to clarify several key points to demonstrate the practical feasibility and memory efficiency of our method.
> - Firstly, as TokenGuard is a multi-stage framework method for controlling hallucinations, information in the form of hidden states or vector representations needs to be passed between stages. To **reduce memory overhead**, most of the information we pass uses temporary space for storage and is **deleted promptly** when no longer needed.
> - Specifically, the hidden state of a token is kept in program memory when it is selected. Therefore, when using the token-level hidden state to calculate the average value, it can be **called directly from memory**. This **solves the problem of doubling the hidden state memory** required when calculating the average value. After the target segment is accepted or discarded, the temporary token hidden state is immediately released, retaining only the compressed segment vector $H_k$.
> - To improve the clarity of the memory usage process, we have revised Chapter 4 to explicitly describe the memory management strategy (**Memory Note**) after each stage's subsection. We have also added a detailed stage-by-stage memory analysis table below in main paper Section5.7 to quantify and compare the footprint of our method.
>
> | Method       |Token(%) | Segment(%) | Global (%) | Peak (%) | Avg (%) | Runtime(s)
> |-----------|----------|--------------|---------|-----------|-----------|-----------|
> | Guided Decoding           | -              | -               | -               | 21.4           | ~15.0          |77.31
> | Chain-of-Thoughts           | -              | -               | -               | 21.3           | ~14.5          |7.02
> | Tree-of-Thought          | -              | -               | -               | 21.4           | ~17.8          |35.04
> | Predictive Decoding           | -              | -               | -               | 4.1           | ~3.7          |58.26
> | Token-Guard  | 1.2 – 9.4    | 21.6 - 19.6             | 10.3 – 19.0     | 21.6           | ~16.3          |53.89
>
> ---
>
> **W4: "In Line 259, the paper mentions the use of an external verification score E_k in order to compute the factual consistency score. Could the authors kindly clarify if an external score is actually utilized for the Stage 3 global iteration? This is an extremely pertinent aspect, since the baseline comparisons would be very different once external verification tools can be incorporated. Additional details on the computation of the sim_ctx score which quantifies semantic alignment is also needed."**
> - We thank the reviewers for the insightful questions regarding the use of external verification and the computation of the $\text{sim}_{\rm ctx}$ score.
> - TokenGuard aims to improve the reliability of large models generating answers within contextual situations through multi-level hallucination detection, **without introducing external tools or additional training**.
> - Firstly, as the reviewers pointed out, the term "external validation score" may be confusing. In our manuscript, we originally defined it as **an external validation score $E_k$ that compares generated content with relevant knowledge sources.** To avoid ambiguity, we will replace "external validation score" with **input-based consistency score**, which more accurately reflects its source.
> - Moreover, we clarify that none of the baseline comparison experiments used external validators. All main table results were evaluated **entirely using internal signals**, so the improvements observed in TokenGuard cannot be attributed to external resources or additional supervision. This ensures a fully fair comparison.
>
> - $sim_{ctx}$ measures the semantic alignment between two adjacent segments by comparing their average token embeddings.
> While hidden-state similarity reflects internal continuity, $sim_{ctx}$ captures text-level relatedness and provides a complementary, more stable signal.**The specific calculation formulas are given in the corresponding section of the main text.**

---

> ### Author Response · Authors · 2025-11-21
> **Response to Reviewer ATsM (Part 3/4)**
>
> ---
> **W5: "Several scoring methods introduced seem poorly motivated. For instance, the convex combination of the cosine similarity of hidden embeddings and the final probability of a given output token (Eq. 5) does not seem well-principled a priori. More clarity is needed on which layer is chosen and how this influences the score. Prior works such as INsiDE [L] use similar components but are more principled and well-motivated."**
>
> - We sincerely thank the reviewer for the insightful and constructive feedback. We will learn from INsiDE and provide a clearer exposition of our method motivation.
> - TokenGuard’s primary goal is to enhance the factual reliability of large-model generation by detecting and mitigating hallucinations. In self-consistent decoding, relying solely on either output probability or internal representation consistency is insufficient; **combining probabilistic and representational cues improves factual reliability.** Accordingly, Eq. (5) employs a convex combination of the token probability and the cosine similarity of hidden embeddings, providing an efficient way to **integrate these signals** within **a single decoding pass** without requiring multiple generations.
>
> - For the cosine similarity, we compute embeddings from the **penultimate transformer layer (L−1)**, which retains richer semantic structure than the final layer, which often undergoes task-specific projection that reduces representational diversity. For TokenGuard, this choice is critical: **L−1 embeddings provide stable, informative token-level representations** that can be reliably aggregated into segment vectors, enabling **robust multi-level hallucination scoring**.
>
> - In the revised manuscript, we have **expanded Section 3.2** to clarify both the motivation for Eq.(5) and the rationale for selecting the penultimate layer, explicitly connecting these design choices to TokenGuard’s practical objectives.
>
> ---
>
> **W6:"For the final stage, with alobaliteration and corrections, could the authors clarify how many chains are used? Why not just use the original order ofthe seqments?"**
>
> - We thank the reviewer for raising these important questions, we will provide further details and design motivation about the final stage of our method.
>
> - The core objective of Token-Guard is to reduce hallucinations in **real-time question answering**. In the global iteration and correction stage, the number of candidate chains **depends on the task scale**: for large-scale datasets, we set the number of K-Means clusters to $K=5$, whereas for smaller datasets such as HaluEval, we use $K=3$ to reduce computational cost and **maintain efficient answer generation**.
> - We do not rely solely on the original segment order because exploring **multiple candidate chains** helps the model to produce **globally consistent** and **factually accurate** answers. In our experiments, we observed that when the input text contains **loosely connected segments**, multiple reasoning paths can achieve higher logical consistency. This design is also consistent with prior work on **branch-based reasoning** "Branch-Solve-Merge Improves Large Language Model Evaluation and Generation", which shows that multiple reasoning paths generally outperform single-sequence reasoning in overall answer quality.
> - For full transparency, we will explicitly report the candidate-chain range (3–5) and our design motivation for multiple reasoning paths in **Section 3.3**.

---

> ### Author Response · Authors · 2025-11-21
> **Response to Reviewer ATsM (Part 4/4)**
>
> **W7: Given the correction and verification steps, comparisons with prior works such as SELF-REFINE, Self-Reflection, Self-Correct, Self-RAG would be highly relevant baselines to compare and contrast with.**
>
> - We thank the reviewer for highlighting these **valuable baseline methods. SELF-REFINE, Self-Reflection, Self-Correct, and Self-RAG** all represent important lines of work in iterative refinement and self-correction. In the final version, we have included all the baselines across all seven evaluation datasets, with the only exception of Self-Correct, whose authors explicitly state in their arXiv version that the code is not publicly available.
>
> | Method             | FinanceBench | RAGTruth | covidQA | DROP_history | DROP_nfl | PubmedQA | Halueval | Avg   |
> |--------------------|--------------|----------|---------|--------------|----------|----------|----------|-------|
> | BaseModel          | 16.00        | 14.71    | 32.33   | 44.21        | 39.10    | 9.55     | 42.16    | 28.29 |
> | Self-RAG           | 42.74        | 32.91    | 42.48   | 50.13        | 45.29    | 43.21    | 86.70    | 51.78 |
> | SELF-REFINE        | 17.09            | 28.93        | 38.66       | 42.64            | 47.25        | 19.47        | 71.75        | 37.97     |
> | Self-Reflection    | 19.37            | 14.82        | 33.76       | 47.48            | 42.05        | 12.83        | 46.21        | 30.93     |
> | **Token-Guard (Ours)** | **30.80**        | **43.94**    | **47.64**   | **68.52**        | **58.10**    | **29.67**    | **78.54**    | **51.03** |
>
> - From the table, we can observe that SELF-REFINE and Self-RAG show strong improvements on short-text or low-complexity tasks. However, their improvements **diminish on long-context, high-compositionality datasets** such as DROP and CovidQA. In particular, **Self-RAG heavily depends on the quality of the constructed evidence corpus**, resulting in **steep performance drops** when retrieval becomes unreliable in long-context settings. **Self-Reflection also shows limited gains**, as multiple rounds of regeneration often **fail to effectively reduce the gap between the predicted answer and the gold answer**, leading to minimal F1 improvement.
> - In contrast, Token-Guard maintains **robust improvements across both short and long contexts**, without requiring retrieval, external evidence, or corpus construction. As a result, **Token-Guard achieves substantial absolute F1 improvements** even on the most challenging datasets.
> - We have **added the relevant experimental results to Appendix D “Comprehensive Evaluation”**. Incorporating these additional baselines substantially **strengthens the reliability of the results** and provides a more comprehensive perspective on iterative refinement and self-correction approaches.
> ---
>
> **Q2:Line 189–194: The difference between the hidden state $ h_t^{(i)} $ and $ H_t^{(i)} $—which denotes the representation vector of the candidate token—was not apparent. Could the authors kindly clarify this?**
>
> - We thank the reviewer for the comment. To clarify, the correct notation for the representation vector of the candidate token is $ h_t^{(i)} $; we have updated the manuscript accordingly.
>
> ---
> **Minor Typos**
> - We thank the reviewers for the careful observations and constructive suggestions. All identified typographical errors have been corrected in the revised manuscript.
>
> ---
> At last, we sincerely appreciate your valuable feedback. We have carefully considered all your suggestions and substantially improved the paper accordingly. If our responses and revisions address your concerns, we would be deeply grateful if you could kindly reconsider raising the score to 6 or above. Thank you very much!

---

### Official Review · Reviewer_eh4g · 2025-11-01

**Soundness:** 2
**Presentation:** 2
**Contribution:** 2
**Rating:** 4
**Confidence:** 4

**Summary:**

This paper presents Token-Guard, a decoding method to mitigate the hallucination through token level self-checking and iterative refinement. Token-Guard integrates three key components: (1) token-level hallucination scoring using latent-space similarity and probability weighting (2) segment-level scoring combining local consistency and global alignment and (3) global iteration with factual and logical verification across reasoning chains. Token-Guard prunes low-confidence tokens and segments during generation, aiming to prevent error propagation. Experiments across six benchmarks demonstrate improved factuality and consistency over four decoding baselines.

**Strengths:**

1. The paper defines a well-structured three-stage pipeline, incorporating token-level, segment-level, and global iteration.
2. The experiment spans six benchmarks, demonstrating remarkable performance compared with other baselines.
3. The method solves a meaningful task. It detects and mitigates the hallucination at the token-level, segment-level, and globally, preventing hallucination propagation.
4. The case study in Table 4 explicitly demonstrates token-wise correction, supporting the claim of local self-repair.

**Weaknesses:**

1. The paper claims that current decoding methods lack token-level hallucination checking mechanism. But layer contrasting method (e.g. DoLa, Contrastive decoding) already mitigate hallucinations at the token level and achieve strong results. Relevant baselines are not discussed or compared.
2. In the latent token environment initialization, the method mentioned it requires initializing the accepted tokens a_j, but no details are provided.
3. Some datasets (e.g., RAGTruth, PubMedQA) show near-zero EM but high F1 (Table 1), suggesting mismatched evaluation or tokenization inconsistencies, but no explanation was found.
4. The experiment was conducted on limited LLM architectures (llama3.1 and Qwen 3), and model scale(3B and 8B).
5. The paper could benefit from providing a thorough examination of failure modes, especially those failed cases after re-generation.

Typo: Ht^{(i)} in line 194 should be H_t^{(i)}

**Questions:**

1. The method selects tokens that are similar to the previously accepted tokens. Could this reduce generation diversity or lead to over-constrained outputs?
2. Line 323, the author set the softmax temperature to 0.3, and the sampling temperature to 0.4. Is there any specific reason to set such low temperatures?
3. Table 3 lists Time and output token comparison but did not provide a normalized comparison(e.g., token/sec). Providing such results would be more straightforward.
4. Some tables and figures are far from their textual references. Repositioning them near relevant sections would improve readability.

---

> ### Author Response · Authors · 2025-11-21
> **Response to Reviewer eh4g (Part 1/4)**
>
> Thank you very much for your time and effort in reviewing our paper. We sincerely appreciate your feedback. Below, we respectfully provide our detailed responses to address your concerns.
>
> ---
>
> **W1:** The paper claims that existing decoding methods lack token-level hallucination checking, but prior layer-contrastive approaches (e.g., DoLa, Contrastive Decoding) already perform token-level mitigation. These relevant baselines are not discussed or compared.
>
> - We thank the reviewer for this insightful comment.
> - To be precise, our intention was to state that, **to the best of our knowledge, prior works have not introduced an explicit token-level hallucination detection mechanism**. While DoLa and other contrastive-layer methods operate at the token level, their approach is limited to **logit correction**. In contrast, Token-Guard evaluates each candidate token and assigns an explicit hallucination score.
>
> - We have included DoLa as a baseline wherever feasible. Table 1 compares Token-Guard and DoLa on the StrategyQA dataset using the LLaMA-7B model:
>
> | Method      | Accuracy (%) | Base Model Accuracy (%) | Improved Accuracy (%) | Improvement Rate (%) |
> | :---------- | :---------- | :-------------------- | :-------------------- | :------------------- |
> | DoLa        | 64.1         | 60.1                    | 4.0                   | 6.66                 |
> | Token-Guard | 69.4         | 62.3                    | 7.1                   | 11.40                |
>
> - In the StrategyQA test, Token-Guard's refusal-to-answer rate was 6.1%, yielding a true error rate of only 24.5%, which demonstrates its effectiveness.
>
> -  We have **clarified the original description in the introduction part** and **included the experimental results and analysis in Appendix~[Open-Domain QA Results]** to provide a more precise presentation and further demonstrate Token-Guard's potential on open-domain QA datasets.
>
> ---
>
> **W2:** In the latent token environment initialization, the method states that it requires initializing the accepted tokens ($a_j$), but no implementation details or initialization procedure is provided.
>
> - We thank the reviewer for the detailed and constructive feedback. We will address implementation details below:
>
> - At generation step $t$, the latent environment $S_t$ stores the semantic representations $s_j$ and contextual hidden states $h_j$ of all previously accepted tokens $a_j$ for $j < t$. To handle the initialization at $t = 1$, when no accepted tokens are available, we introduce an anchor state derived from the input context. Specifically, we compute the mean penultimate-layer hidden state of the input prompt $x = (x_1, \dots, x_{|x|})$:
> $h_x$ is obtained by taking the average of the hidden states (from layer $L-1$) of all tokens in $x$.
> We set $h_{<1}$ equal to this averaged vector. **The specific calculation formulas are given in the Section 4.2.**
> - Here, $\mathrm{LLM}_{\text{hidden}}^{(L-1)}$ denotes the hidden representation from the second-to-last layer of the language model. This initialization ensures that trajectory coherence and hallucination scoring are well-defined from the very first generated token, avoiding edge cases caused by an empty history. Importantly, $h_x$ and all subsequent token hidden states $h_j$ reside in the same vector space, enabling valid similarity computations throughout generation.
>
> - We have **updated the manuscript to include this explicit initialization procedure in Section 4.1** and we hope this clarification addresses the reviewer’s concern.
>
> ---
>
> **W3:** Some datasets (e.g., RAGTruth, PubMedQA) show near-zero EM but high F1 scores (Table 1), suggesting evaluation mismatch or tokenization inconsistencies, but the paper does not provide explanations.
>
> - We thank the reviewer for this insightful and important observation. We address this concern below:
>
> - Token-Guard is designed to mitigate hallucinations in context-dependent question answering tasks. To evaluate its robustness, we incorporate **short-context datasets** such as DROP and HaluEval, which offer well-defined question–answer pairs, as well as **long-context datasets** including RAGTruth and PubMedQA, whose answers are expressed in the form of **explanatory long-form text**.
>
> - As described in the **Appendix[Evaluation Details]**, **Exact Match requires the predicted answer to replicate the reference string precisely**, whereas the F1 score reflects the degree of token-level overlap between the two. For RAGTruth and PubMedQA, very low Exact Match values naturally occur due to the **length and structural variability** of the answers. This pattern reflects the characteristics of the task rather than an issue in evaluation.
> - To maintain fairness across datasets, all experiments use **consistent tokenization procedures**.
> - We have **revised the manuscript in 5.2 "Main Results (RQ1)**" to clearly explain this evaluation mismatch and to clarify the intended use and interpretation of Exact Match and F1 in our study.

---

> ### Author Response · Authors · 2025-11-21
> **Response to Reviewer eh4g (Part 2/4)**
>
> ---
>
> **W4:** Experiments were conducted on limited LLM families (Llama 3.1 and Qwen 3) with relatively small scales (3B and 8B), which limits generalizability.
>
>
> - We sincerely thank the reviewer for raising this important point regarding model coverage and generalizability.
> - Our experiments primarily focus on 8B-class models, which are widely used in **resource constrained** applications. Both Llama-3-8B and Qwen2-7B/8B are representative open-weight architectures and cover **the majority of QA deployment scenarios**. We additionally evaluate 3B-scale models to **examine stability and robustness** under limited compute budgets, demonstrating that Token-Guard remains reliable even with smaller backbones.
> - To further address the reviewer's concerns regarding model scale, we include additional experiments on **13B-class models** using **Llama-2-13B**. Predictive Decoding (PD) is omitted because it consistently triggered out-of-memory errors during decoding. The results show that Token-Guard continues to **deliver substantial improvements** over all baselines, suggesting that its core mechanisms generalize well across 3B, 8B, and 13B model families.
> Below we provide the complete results on the 13B backbone:
>
> | 13B Model Results |       |         |              |        |             |       |            |        |
> | :---------------: | :---: | :-----: | :----------: | :----: | :---------: | :---: | :--------: | :----: |
> |    **Method**     | **RAGTruth** |       | **DROP_history** |        | **HaluEval**  |       |            |        |
> |                   |   F1  |  BLEU   |      F1      |  BLEU  |     F1      | BLEU  |            |        |
> |    Base Model     | 22.47 | 15.26   |    46.32     | 42.74  |    35.30    | 31.40 |            |        |
> | Chain-of-Thought  | 32.25 | 27.67   |    33.90     | 30.06  |    49.43    | 44.01 |            |        |
> |  Tree-of-Thought  | 32.93 | 23.70   |    43.97     | 38.25  |    62.20    | 58.07 |            |        |
> | Predictive Decoding |  —  |   —     |      —       |   —    |      —      |   —   |            |        |
> |  Guided Decoding  | 29.34 | 22.01   |    40.41     | 36.71  |    63.42    | 58.72 |            |        |
> | **Token-Guard (Ours)**| **33.14** | **30.41**   |   **51.17**     | **47.64**  |    **72.66**    | **68.81** |            |        |
>
> - We have **updated Section 5.7** of the manuscript to include these results and to clarify our rationale for model selection, emphasizing practical deployment considerations.
>
> ---
>
> **W5:** The paper does not provide a thorough analysis of failure modes, especially for cases that still fail after re-generation.
>
> - We thank the reviewer for highlighting the importance of analyzing residual failure cases.
> - While Token-Guard substantially reduces hallucinations, it does not guarantee perfect factual correctness. Remaining errors typically arise from **ambiguity or incompleteness in context**, **difficulty in long-form or multi-step reasoning**, and **fine-grained numerical reasoning errors**. These factors explain why some generations remain incorrect even after re-generation.
> - To address the reviewer’s suggestion, we have added a dedicated appendix **“Error Case Studies”**. It provides a systematic analysis of representative failure examples across datasets, illustrating why certain cases remain unsolved. We hope this will help the reviewer and readers better understand the **limitations of current LLMs in context-dependent QA tasks** and appreciate the **practical strengths and boundaries** of Token-Guard.
> ---
>
> **W6:** There is a minor typo: “(H_t^i)” in line 194 should be written as “(H_t^{(i)})”.
>
> - We thank the reviewer for the comment. To clarify, the correct notation for the representation vector of the candidate token is $h_t^{(i)}$; we have **updated the manuscript** accordingly.

---

> ### Author Response · Authors · 2025-11-21
> **Response to Reviewer eh4g (Part 3/4)**
>
> ---
>
> **Q1:** The method selects tokens similar to previously accepted tokens. Could this suppress generation diversity or cause overly constrained outputs?
>
> - We sincerely thank the reviewer for raising this important question regarding potential limitations on generative diversity. We would like to clarify the motivation and design of this component.
>
> - Our method employs token-level similarity checks **specifically for hallucination suppression**, aiming to ensure factual consistency and stable reasoning trajectories. The comparison with previously accepted tokens is not intended to restrict expressive diversity, but rather to prevent **local contradictions** and maintain **coherence with the reasoning context**. Each candidate token is evaluated with respect to both contextual alignment and ongoing reasoning consistency, rather than being forced toward a single deterministic path.
>
> - Moreover, **diversity is preserved at the system level** through multiple candidate branches, re-generation steps, and segment-level aggregation, which allow the model to explore alternative reasoning chains while still constraining hallucination drift. This multi-stage structure ensures that the method balances factual reliability with expressive flexibility.
>
> - We hope this clarification illustrates how Token-Guard’s token selection strategy preserves generative richness while ensuring consistency and factual reliability.
> ---
>
> **Q2:** In line 323, the authors set the softmax temperature to 0.3 and the sampling temperature to 0.4. Is there a specific reason for choosing such low temperatures?
>
> - We appreciate the reviewer's question regarding our choice of relatively low softmax temperature (0.3) and sampling temperature (0.4). To clarify this design decision, we provide both **quantitative evidence** and **conceptual explanation** of our temperature selection philosophy.
> - Token-Guard fundamentally relies on **stable token-level hidden-state trajectories** to perform reliable self-checking. Higher temperatures introduce **excessive stochasticity** during decoding and consequently **weaken the consistency scores** that our method depends on. In contrast, **lower temperatures smooth local transition dynamics and preserve the fine-grained signals required for hallucination detection.**
>
> - Our choice is also directly supported by Li et al. (2025)[1] Their study concludes that **low-temperature decoding (typically around 0.4) yields more stable distributional alignment in self-consistency reasoning**. We also conducted temperature comparison experiments, as shown in the table below. The results confirm that **higher temperatures degrade the quality of the consistency signal**, providing clear empirical support for our selected temperature range.
>
> | Softmax / Sampling | HaluEval F1 | RAGTruth F1 |
> | ------------------ | ----------- | ----------- |
> | 0.15 / 0.25        | 75.63       | 41.94       |
> | **0.30 / 0.40**        | **79.56**      | **43.60**       |
> | 0.50 / 0.60        | 73.34       | 43.31       |
> | 0.80 / 0.80        | 67.05       | 42.30       |
> - To further enhance clarity and methodological transparency, we have included an explicit explanation of this design choice in the **Appendix[Selection of Temperatures]**, ensuring that readers clearly understand the motivation behind our use of the 0.3–0.4 temperature range. We hope this resolves the reviewer's concerns.
>
> ---
>
> [1] Revisiting Self-Consistency from Dynamic Distributional Alignment Perspective on Answer Aggregation, Li et al., arXiv 2025

---

> > ### Author Response · Authors · 2025-11-21
> > **Response to Reviewer eh4g (Part 4/4)**
> >
> > ---
> >
> > **Q3:** Table 3 reports time and output token counts, but does not provide normalized throughput metrics (e.g., tokens/sec). Would including such normalized results provide clearer comparison?
> >
> > - We thank the reviewer for the insightful suggestion. We agree that normalized throughput metrics (e.g., tokens/sec) can provide a clearer and more comparable view of system efficiency across different decoding methods.
> > - In the revised version, we have added a normalized throughput column for each dataset and included an averaged summary. This additional metric highlights the relative decoding efficiency more transparently. The updated table is provided below and has been integrated into the main paper，Table 3.
> >
> >
> > | **Method**              | **RAGTruth**             | **covidQA**              | **PubmedQA**             | **HaluEval**             | **Avg.**                 |
> > |------------------------|---------------------------|---------------------------|---------------------------|---------------------------|---------------------------|
> > |                        | Time · Tokens · Tokens/sec | Time · Tokens · Tokens/sec | Time · Tokens · Tokens/sec | Time · Tokens · Tokens/sec | Time · Tokens · Tokens/sec |
> > | **Chain-of-Thoughts**  | 11 · 899 · 80.71          | 9 · 670 · 73.14           | 11 · 998 · 88.45          | 7 · 543 · 77.33           | 9.65 · 777.08 · 79.41     |
> > | **Tree-of-Thought**    | 57 · 3687 · 64.49         | 67 · 3423 · 51.09         | 58 · 4253 · 73.38         | 35 · 2786 · 79.49         | 54.29 · 3537.19 · 67.11   |
> > | **Predictive Decoding**| 107 · 12058 · 112.82      | 138 · 6473 · 47.07        | 79 · 8416 · 106.06        | 58 · 5954 · 102.20        | 95.50 · 8225.61 · 92.54   |
> > | **Guided Decoding**    | 110 · 17234 · 155.96      | 189 · 13525 · 71.64       | 103 · 18733 · 182.05      | 77 · 11689 · 151.15       | 119.38 · 15295.41 · 140.20 |
> > | **Token-Guard** | 69 · 18024 · 262.81 | 301 · 17474 · 58.10 | 32 · 7699 · 240.58 | 54 · 5254 · 97.54 | 113.29 · 12112.95 · 164.76 |
> >
> > - As shown in table, Token-Guard achieves the **highest average throughput** among all methods, showing that its multi-stage hallucination control **introduces minimal decoding overhead** while avoiding the heavy branching costs of ToT and PD. This demonstrates that Token-Guard is both **more efficient and more effective**, producing cleaner reasoning with fewer redundant tokens.
> >
> > ---
> > **Q4:** Some tables and figures are placed far from their textual references. Would repositioning them closer to the relevant discussion improve readability?
> >
> > - We appreciate the reviewer’s suggestion. We agree that placing tables and figures closer to their textual references can improve readability.
> > - Accordingly, **we have repositioned key tables and figures to appear immediately after the relevant discussion**. This adjustment helps readers more easily follow the experimental results and associated analyses.
> >
> > ---
> > At last, we sincerely appreciate your valuable feedback. We have carefully considered all your suggestions and substantially improved the paper accordingly. If our responses and revisions address your concerns, we would be deeply grateful if you could kindly reconsider raising the score to 6 or above. Thank you very much!

---

> > > ### Comment · Reviewer_eh4g · 2025-11-26
> > > **Response to the rebuttal**
> > >
> > > Thank you for the responses. I appreciate the efforts on the additional experiments. My main concerns have been resolved. Therefore I raise my score to 6.

---

### Author Response · Authors · 2025-11-29
**Author Final Remarks**

Dear PCs, SACs, ACs,and Reviewers，

We thank all reviewers for their valuable feedback, Below we summarize each reviewer's main concerns and our responses.

**eh4g:**
- *Concerns*: Missing layer-contrastive baselines, unclear latent initialization, long-form mismatch, limited model scale, missing failure modes, typos, diversity suppression, temperature rationale, lack of normalized throughput, and layout issues.
- *Response*: We add DoLa on StrategyQA(Section 2 & Appendix D), align long-form evaluation, analyze failures(Appendix I), fix typos, report normalized throughput(Table 4), and reorganize tables/figures. Token-Guard scores tokens directly, initializes latent states with hidden-mean, scales from 3B–13B(Section 5.8), preserves diversity via multi-branch regeneration, and stabilizes decoding with low temperature. All clarifications are reflected in the revised paper and appendix.
- *Addition*：**After reviewing our rebuttal, the reviewer decided to raise the score from 4 to 6 at 1:29 PM on November 25 (AoE)**.

**ATsM:**
- *Concerns*: Hyperparameter practicality, early-segment refinement, memory cost, external verification, sim_ctx computation, scoring/layer motivation, global iteration, missing baselines, notation issues, and typos.

- *Response*:  We use a unified grid-search for all datasets/models, refine only low-confidence segments to keep states stable, manage memory with temporary hidden states and compressed vectors, and avoid external tools. We clarify sim_ctx, convex scoring with L−1 embeddings, and the logic behind multi-path global iteration (3–5 chains). We add SELF-REFINE, Self-Reflection, and Self-RAG, showing strong gains (Appendix D.1). All technical details are updated across the main paper and appendix.


**35wE:**
- *Concerns*: Lightweight-claim clarity, hyperparameter robustness, latent-score validity, generality, efficiency, and score visualization.
- *Response*: We integrate Token-Guard into standard decoding without extra training, verify hyperparameter robustness via grid search, and use cosine-based latent scoring for stable multi-stage detection. Our evaluation covers numerical, multi-hop, domain QA, and StrategyQA. We include efficiency analysis and hallucination-score visualizations(Section 5.7), and refine terminology, notation, and baselines. These revisions are incorporated into the updated manuscript.

**5cxV:**
- *Concerns*: Multi-layer decoding efficiency, adaptive hyperparameters, RLHF/RAG comparison, fragmented inputs, edge-case hallucinations, and unclear method description.

- *Response*: We maintain high throughput with fixed hyperparameters, add minimal overhead, and require no retrieval or extra training. Token-Guard remains robust across short/long contexts and mitigates decoding-induced hallucinations(Self-RAG comparison). We cite prior token-level detection work (Section 2), and release all scoring/memory details. These updates are included in the revised paper.

**In summary, as of 1:29 PM on November 25 (AoE), our overall score increased from 8644 to 8664.**

We believe these responses effectively address all concerns, demonstrating Token-Guard's robustness, fairness, and scalability.

Best,


The Token-Guard Authors

---

### Meta-Review · Area_Chair_KXZd · 2026-01-06

**Summary:**

This work studies token-level hallucination control of LLMs. The authors propose Token-Guard, which refines low-confidence tokens and segments during decoding. Token-Guard enjoys several favourable properties, such as training-free, hierarchical hallucination control, and significant empirical improvements.

Reviewers share concerns regarding the complexity of the method, hyperparameter sensitivity, efficiency, and comparison with related works. The authors’ rebuttal provides additional experiments and clarifications that address the main concerns. The additional discussion with related works, empirical comparison with existing RAG methods, efficiency studies, and analysis will further strengthen this work.

The authors are encouraged to incorporate all promised revisions and further improve clarity in the revised version, especially around the hyperparameter selection protocol and the exact scope/assumptions of the refinement procedure.

**Reviewer Concerns:**

Reviewers initially raised several concerns regarding:
- Complexity of the method: The multi-stage decoding pipeline requires careful control or even some heuristics, and the presentation of the method does not clearly state some important details (ATsM, 35wE). With explanation and more efforts in improving the clarity of this work, this concern can be partially addressed, while it does not affect the nature of the relatively complex control of the decoding process.
- Hyperparameter sensitivity: There are many hyperparameters in Token-Guard (e.g., thresholds, weights, iteration limits) to be tuned (ATsM, 35wE, 5cxV). Regarding this concern, the authors present a detailed hyperparameter robustness study and also note that the hyperparameter of Token-Guard is fixed across all benchmarks. The results could serve as strong evidence to mostly address this concern.
- Efficiency: The complex decoding strategy may also require more runtime and memory overhead (eh4g, 5cxV, 35wE). The authors provide direct evidence to address this concern.
- Comparison with related works: For example, Reviewer eh4g highlighted the comparison between another "token-level" mitigation strategy, i.e., DoLa/layer-contrastive decoding; Reviewer ATsM and 5cxV also suggested comparison with self-refine, self-reflection, and self-rag methods, as they share the same nature of self-checking as Token-Guard. The authors also provide direct responses and promise revisions to address this concern.
- Limited evaluation: Experiments with more models, benchmarks, and failure mode analysis are suggested (eh4g, 35wE, 5cxV). The authors follow the reviewers' suggestions and provide results; this concern can also be addressed.

Overall, the main concerns of the reviewers are effectively addressed by the authors.

**Reviewer Scores:**

This work initially received a mixed rating from the reviewers, with two negative scores. After checking all the reviewers' concerns and the authors' response, the majority of the concerns can be addressed by the authors, especially those shared by the reviewers (ATsM,eh4g). As stated in the review comments that they are open to changing the score, the two reviewers are highly likely to increase the score, which makes this work overall positive. Hence, I lean to a positive recommendation.

---

### Decision · Program_Chairs · 2026-01-26

Accept (Poster)